

# Modern calibration of *Poa flabellata* (Tussac grass) as a new paleoclimate proxy in the South Atlantic

Dulcinea V. Groff[1,2], David G. Williams[3], Jacquelyn L. Gill[1,2]

[1]Climate Change Institute, University of Maine, Orono, ME 04469, USA
[2]School of Biology and Ecology, University of Maine, Orono, Me 04469, USA
[3]Department of Botany, University of Wyoming, Laramie, WY 82071, USA

*Correspondence to*: Dulcinea V. Groff (dulcineavgroff@gmail.com)

**Abstract.** Terrestrial paleoclimate records are rare in the South Atlantic, limiting opportunities to provide a prehistoric context for current global changes. The tussock grass, *Poa flabellata*, grows abundantly along the coasts of the
Falkland Islands and other sub-Antarctic islands. It forms extensive peat records, providing a promising opportunity to reconstruct high-resolution regional climate records. The isotopic composition of leaf and root tissues deposited in these peats has the potential to record variation in precipitation, temperature, and relative humidity over time, but these relationships are unknown for *P. flabellata*. Here, we investigate the isotopic composition of *P. flabellata* plants and precipitation and explore seasonal relationships with temperature and humidity across 4 study locations in the Falkland
Islands. We reveal that inter-seasonal differences in carbon and oxygen stable isotopes of leaf α-cellulose of living *P. flabellata* significantly correlated with monthly mean temperature and relative humidity. The carbon isotope composition of leaf α-cellulose ($\delta^{13}C_{leaf}$) records the balance of $CO_2$ supply through stomata and the demand by photosynthesis. The positive correlation between $\delta^{13}C_{leaf}$ and temperature and negative correlation between between $\delta^{13}C_{leaf}$ and relative humidity suggest that photosynthetic demand for $CO_2$ relative to stomatal supply is enhanced
when conditions are warm and dry. Further, the positive correlation between $\delta^{13}C_{leaf}$ and $\delta^{18}O_{leaf}$ (r = 0.88, p < 0.001, *n* = 24) indicates that stomatal closure during warm dry periods explain seasonal variation in $\delta^{13}C_{leaf}$. We observed significant differences between winter and summer seasons for both $\delta^{18}O_{leaf}$ and $\delta^{13}C_{leaf}$, and among study locations for $\delta^{18}O_{leaf}$, but not $\delta^{13}C_{leaf}$. $\delta^{18}O$ values of monthly composite precipitation did not differ between seasons or among study locations, yet is characteristic of the latitudinal origin of storm tracks and seasonal winds. The weak correlation
between $\delta^{18}O$ in monthly composite precipitation and $\delta^{18}O_{leaf}$ further suggests that relative humidity is the main driver of the $\delta^{18}O_{leaf}$. The oxygen isotopes in root α-cellulose did not reflect, or only partially reflected (at one study location), the $\delta^{18}O$ in precipitation. Overall, this study supports the use of peat records formed by *P. flabellata* to fill in a significant gap in our knowledge of the long-term trends in Southern Hemisphere climate dynamics.

## 1 Introduction

The high latitude environments of the South Atlantic are changing rapidly. Over the last century, mean annual temperature in the Falkland Islands (Fig. 1A) has increased by 0.5 ºC (Lister and Jones, 2015). This warming has corresponded with an intensification and poleward shift of the southern westerly winds and aridification (Gillett et al., 2008; Thompson and Solomon, 2002; Villalba et al., 2012). These changes are already altering the distribution of





marine animals in the Southern Ocean (Weimerskirch et al., 2012), and warming of the western South Atlantic is
projected to alter the distribution of island plants as well (Jones et al., 2013; Upson et al., 2016). The inconsistency of
meteorological measurements from the Falkland Islands dating back to 1874 (Lister and Jones, 2015) means we lack
critical information on the long-term patterns and whether these are novel conditions. Paleoecological archives, such
as high-resolution lake sediments and tree rings, can provide useful long-term records documenting and quantifying
changes (Dietl et al., 2015; Dietl and Flessa, 2011; Willis et al., 2010), but such records are lacking for the South
Atlantic. The absence of trees and deep lakes across many sub-Antarctic islands especially limits high-resolution,
independent paleoclimate reconstructions, which are essential for detecting past abrupt climate change. However,
many sub-Antarctic islands support widespread communities of peat-forming C3 tussock grasses (*Poa flabellata*),
which provide important habitat and shelter for breeding marine animals such as seals and seabirds. Peat records
formed by *P. flabellata* present a promising avenue for paleoclimate reconstructions; peatland vegetation has been
used to reconstruct hydrological change and temperature in mid- to high latitudes (Amesbury et al., 2015; Chambers
et al., 2012; Pendall et al., 2001).

*P. flabellata* grasslands in the South Atlantic accumulate substantial amounts of peat (Smith and Clymo, 1984), and
have the highest carbon accumulation rates of any peatland globally (Payne et al., 2019). Endemic to the South
Atlantic, *P. flabellata* only occurs on Tierra del Fuego, the Falkland Islands, Gough Island, and South Georgia. *P.*
*flabellata* grasslands were once widespread throughout the Falklands, but are now greatly reduced because of land-
use change and introduced grazers (Strange et al., 1988; Wilson et al., 1993). The term "tussock" is used to describe
the clumping growth form of *P. flabellata,* while the species itself is commonly known as "tussac."

Several factors support the utility of *P. flabellata* peats as a paleoclimate proxy. *P. flabellata* peatlands are formed by
a nearly single-species community of *P. flabellata,* which allow very little light or space for other plants to co-occur
in the absence of disturbance. Tillers of *P. flabellata* grow on top of a pedestal of decaying roots and leaves (called a
"bog") that can reach 4 meters high (Fig. 2A; (Smith and Clymo, 1984); mature plants thus likely primarily use water
from precipitation, as they are not rooted in the soil directly. (Smith and Prince, 1985) established radiocarbon ($^{14}$C)
dates for a *P. flabellata* pedestal and estimated an age of 250 to 330 years. *P. flabellata* grass forms extensive peat
deposits of up to 13.3-m deep, with carbon accumulation rates of 139 g C m$^{-2}$ yr$^{-1}$ (Payne et al., 2019; Smith and
Clymo, 1984), far greater than peatlands of similar latitude in the Northern Hemisphere (18.6 g C m$^{-2}$ yr$^{-1}$), the tropics
(12.8 g C m$^{-2}$ yr$^{-1}$) or Patagonia (22 g C m$^{-2}$ yr$^{-1}$) (Yu et al., 2010). Subfossil *P. flabellata* leaves are abundant in these
peats (Fig. 2E), and readily can be separated from root subfossils. Having the highest accumulation rate of any global
peatlands, *P. flabellata* peat is ideal for high-resolution climate reconstructions. Basal $^{14}$C radiocarbon dates indicate
most *P. flabellata* peatlands initiated between ~ 12,500 and 5,500 $^{14}$C years (Groff, 2018; Payne et al., 2019; Smith
and Clymo, 1984).

Grasses exhibiting the tussock growth form often have evergreen leaves and exhibit a profligate/opportunistic water
use strategy, due to the high evaporative conditions and pulses of water availability in semi-arid habitats (Moreno-
Gutiérrez et al., 2012; Sala et al., 1989; Schwinning and Ehleringer, 2001). The growth phenology of *P. flabellata* is
such that it mainly increases in height in summer (39 cm per year) and in winter an increase in basal area occurs with



the production of new tillers are produced at the base of a 'bog' (Stanworth and Upson, 2013). The climate signal recorded in the cellulose of plant tissues (roots, shoots, and leaves) is deciphered using stable isotopes $\delta^{18}O$, $\delta D$ (Araguás-Araguás et al., 2000) and $\delta^{13}C$. Carbon and oxygen stable isotope ratios record species' water-use strategies in water-limited environments because of physiological responses such as stomatal conductance and assimilation rates (Farquhar and Sharkey, 1982; Moreno-Gutiérrez et al., 2012). Tussock grasses typically occur in water-limited

environments where low water-use efficiency and high stomatal conductance are common functional traits that allow tussock grasses to take advantage of pulses of water (Moreno-Gutiérrez et al., 2012). Correlations between $\delta^{18}O$ of plant cellulose and air temperature and humidity provide information about environmental conditions in the season the cellulose tissue is formed. The $\delta^{18}O$ of leaf water is a primary driver of $\delta^{18}O$ in leaf cellulose, and is influenced by the $\delta^{18}O$ value of plant source water, temperature and humidity (Helliker and Ehleringer, 2002; Roden and Ehleringer,

1999). $\delta^{18}O$ of source water often is correlated with temperature of the environment (Libby et al., 1976). Apart from water source, $\delta^{18}O$ of cellulose also can be influenced by internal exchange among organic molecules and other plant waters (Sternberg et al., 1986). The $\delta^{13}C$ value of leaf biomass in C3 plants records $\delta^{13}C$ of source $CO_2$ and the expression of fractionation effects associated with $CO_2$ diffusion into and through leaf tissue and carboxylation (Farquhar et al., 1982). The net discrimination against $^{13}C$ during photosynthesis is driven by changes in the supply of

$CO_2$ through stomatal pores and demand for $CO_2$ by photosynthetic biochemistry (Cernusak et al., 2013; Farquhar et al., 1982; Ferrio and Voltas, 2005); Fig. 2B). The $\delta^{13}C$ value of roots tends to be 1-3 per mil higher than that of leaves due to a number of post-photosynthetic biochemical fractionations and C allocation pathways (Cernusak et al., 2009).

Plant species vary in the way they isotopically record precipitation and temperature; therefore, peat comprised of a single species is more desirable over a mixture of species (van Geel and Middeldorp, 1988). To test the potential of

*P. flabellata* peatlands as a paleoclimate proxy, we conducted a modern calibration study. We measured oxygen ($\delta^{18}O$) and carbon ($\delta^{13}C$) stable isotopes from living *P. flabellata* leaf tissues (α-cellulose) collected monthly at four sites across the Falklands (51° S, 59° W; Fig. 1). We aim to improve our understanding of Southern Hemisphere westerly wind dynamics with a new paleoclimate proxy that leverages the unique properties of *P. flabellata*.

## 2 Materials and Methods

**2.1 Study location description**

The Falklands (Fig. 1A) are located approximately 500 km east of southern South America, between 51°0.5' S to 52°28.0' S and 61°22.0' W to 57°40.5' W. The cool-temperate (mean temperature: January 9.4 °C and July 2.2 °C) climate of the Falklands is driven by the cold Antarctic Circumpolar Current, the waters surrounding the Antarctic Peninsula, the Falklands Current, and the Andes of southern Patagonia to the west (Turner and Pendelbury, 2000).

The persistent winds of the southwesterly wind belt average 8.5 m s$^{-1}$, with gale force winds averaging 70 days per year and annual precipitation generally ranges between 400 to 600 mm (Jones et al., 2013; Lister and Jones, 2015). Study sites were selected to reflect 1) climatic diversity, and 2) the availability of volunteers to collect monthly samples



for one year. We ultimately selected four sites (Fig. 1B-C): Bleaker Island, Cape Dolphin, Surf Bay, and West Point Island.

## 2.2 Precipitation, temperature and relative humidity

Precipitation was collected at each site using a Palmex monthly composite collection sampler (Palmex d.o.o., Zagbreb, Croatia). The Palmex collector is designed to prevent evaporation and evaporative enrichment of $^{18}O$ in precipitation samples without the use of paraffin oil (Gröning et al., 2012), and has been recommended by the Global Network of Isotopes in Precipitation (GNIP). Samples were shipped to the University of Maine prior to analysis. Oxygen ($\delta^{18}O$) and hydrogen ($\delta D$) stable isotope ratios of water samples were measured at the University of Wyoming Stable Isotope Facility (UWSIF) using a high-temperature conversion elemental analyzer (TCEA) connected to a Thermo Scientific Delta V Plus that is run in continuous flow mode via a ConFlo IV. The technique used injections of 1 µl of water into the TCEA column filled with glassy carbon heated to 1420 °C. Precipitation samples were extracted using cryogenic vacuum distillation (Ehleringer and Osmond, 1989) prior to TCEA analysis to remove aeolian debris, including marine salts. Internal QA/QC working standards calibrated against IAEA international standards Vienna Standard Mean Ocean Water (VSMOW) and Standard Light Antarctic Precipitation (SLAP) and spanning the range of measured values in our study were analyzed with each batch of samples with analytical precision typically better than 0.3 and 2.5 ‰ for $\delta^{18}O$ and $\delta D$, respectively. Isotope values are reported with respect to VSMOW in parts per thousand (per mil, ‰). Explanations of methods for daily average temperature and relative humidity measurements, as well as seasonal wind speed, wind direction, and back trajectory models to determine origins of air masses are found in the Supplemental Text 1.

## 2.3 *Poa flabellata* field collection and cellulose extraction, and isotope analyses

*Poa flabellata* plants were collected at the start of each month at each site from October 2015 through September 2016, from relatively uniform habitats that were undisturbed by grazing or tilling. Up to six *P. flabellata* plant tillers (leaves, stem, and roots) were collected near each of the four stations each month. Whole plants were stored in paper envelopes stored in a cool, dark, dry location until frozen. Samples collected between September 2015 to February 2016 were frozen in February 2016 and samples collected in March 2016 to August/September 2016 were frozen in August/September 2016. Samples were frozen for eight days at the Falkland Islands Department of Agriculture. For leaf material, the inner developing (youngest) leaves were collected. There was no indication that leaves were morphologically different between summer and winter. Only coarse roots were used and fine roots were excluded. Variation in environmental conditions during the growth of a leaf blade can lead to isotopic variations along the gradient of a single leaf as has been shown with $\delta^{18}O$ of cellulose (Helliker and Ehleringer, 2000, 2002); therefore, whole-leaf plant samples were homogenized by drying at 50 °C and pulverizing using a Retsch ball mill at the University of Maine. For each sample, we used 20 mg of pulverized and homogenized leaf or root material for extraction and purification of α-cellulose, following an adapted procedure of (Brendel et al., 2000). Samples were vortexed throughout extraction and purification for homogenization and were visually inspected for purity. Further indicators of purity include undetectable amounts of % nitrogen, and analysis of % carbon in cellulose. As an internal





quality control, one leaf sample was selected for extraction and purification of α-cellulose throughout the sample

processing in batches of 10 to 12 samples. The $\delta^{13}C$ and $\delta^{18}O$ of leaf cellulose for the internal quality control samples

varied by $\pm < 0.1$ ‰ and $\pm 0.3$ ‰, respectively.

Oxygen ($\delta^{18}O$) and carbon ($\delta^{13}C$) stable isotope ratios of α-cellulose samples were measured at the University of

Wyoming Stable Isotope Facility (UWSIF). Oxygen was analyzed using a TCEA coupled to a Thermo Delta V IRMS;

$\delta^{18}O$ values are expressed with regard to VSMOW in parts per thousand (per mil) (Craig, 1961; Gonfiantini, 1978).

Values were normalized to the VSMOW scale using USGS-42 and IAEA-601 cellulose quality control standard

reference materials for oxygen isotopic composition. Analytical precision was $\pm 0.3$ ‰ for $\delta^{18}O$ based on repeated

analysis of internal standards, and samples loaded into silver capsules had weights ranging from 0.206 to 0.289 mg.

Carbon isotope composition of the cellulose samples were determined using a Costech 4010 Elemental Analyzer

coupled to a Thermo Delta Plus XP-IRMS; units are expressed with regard to VPDB in parts per thousand (per mil).

Analytical precision was $\pm 0.1$ ‰ for $\delta^{13}C$ based on repeated internal standards. Quality control standard reference

material for carbon isotopic composition included USGS-40 glutamic acid, USGS-41 glutamic acid, and internal

UWSIF α-cellulose. Carbon sample weights ranged from 1.937 to 2.194 mg and were loaded into tin capsules.

### 2.4 Statistical analysis

For both $\delta^{18}O$ and $\delta^{13}C$, we analyzed the average of three to four plant leaf samples per month for summer (DJF) and

winter (JJA) season at each of the four sites, and the average of up to eight plant root samples (Supplementary Data 1

and 2). We used Pearson's correlation coefficient, r, to detect associations between $\delta^{18}O$ of cellulose and precipitation

samples to test whether $\delta^{18}O$ samples reflects the isotopic value of precipitation. Using Pearson's correlation

coefficient, we tested for a relationship between $\delta^{13}C$ and $\delta^{18}O$ values of cellulose, temperature, and relative humidity.

We tested for a significant difference between summer and winter $\delta^{18}O$ and $\delta D$ in precipitation using a t-test ($n = 24$).

A separate one-way analysis of variance (ANOVA) to test for significant differences among sites included $\delta^{18}O$ and

$\delta D$ in precipitation samples from the entire year ($n = 47$).

A two-way ANOVA compared the main effects of season (summer vs. winter) and the four study locations on the

carbon and oxygen stable isotopes of α-cellulose of *P. flabellata* leaves and roots grown in the summer versus winter,

followed by a post hoc test (Tukey's multiple comparison of means). P-values < 0.05 are considered significant.

Descriptive and multivariate analyses were conducted with SigmaPlot 12.5.

### 3 Results

#### 3.1 Environmental measurements

Across all sites, summer (DJF) daily average temperatures ranged from 3.5 °C to 15.6 °C (mean = 10.0 °C) and relative

humidity ranged from 64.2 % to 98.1 % (mean = 81.1 %). Winter (JJA) daily average temperatures ranged from -1.8

°C to 7.6 °C (mean = 3.7 °C), and relative humidity ranged from 73.6 % to 100 % (mean 94.3 %). Seasonal temperature



(°C) and relative humidity (%) minimum and maximum ranges for individual study locations are found in Table S1. Between study locations, the daily average temperatures over the year ($F_{(3, 44)} = 0.316$, $p = 0.813$, Fig. 2C) and relative humidity were not significantly different ($F_{(3, 44)} = 0.674$, $p = 0.573$, Fig. 2D).

### 3.2 Wind

The wind rose (Fig. S2) shows that winter winds at Bleaker Island primarily blew from the west and northwest. In
winter (JJA), two spokes in the west and NNW direction comprise >30% of the total recorded 15-minute wind directions. In summer (DJF), three spokes in the west, WSW, and SW directions comprise >45% of all 15-minute wind directions. The wind rarely blew from the east, SE, or north. Examining winds from the west in winter, >10% of wind speeds recorded were between 5 and 10 m s$^{-1}$, and the frequency of strongest winds came from the NNE. In summer, >20% winds from the SW were between 5 and 10 m s$^{-1}$, and there was a higher frequency of 10-15 m s$^{-1}$
wind speeds than in winter. Seasonal wind variation deviated from the long-term average (1979-2015). Reanalysis data (ERA Interim; Fig. S3) indicated that the wind speeds during summer (DJF 2015 to 2016) were stronger over the Falklands (5 to 6 m/s) and weaker during winter (JJA 2016).

### 3.3 Seasonal HYSPLIT air mass trajectory analyses

The daily back trajectory HYSPLIT analysis indicated that during the summer, 89% of the air masses originated ($n =$
344) west of the Falklands. Approximately 11% of summer air masses originated south of the Falklands near the Antarctic Peninsula. In winter, the air mass back trajectories ($n = 332$) were from the west, NW, and SW, while 21% of air masses had backward trajectories south of the Falklands near the Antarctic Peninsula (Fig. S4).

### 3.4 Monthly composite precipitation, δ¹⁸O and δD

Each study location had $n = 12$ samples over the year, except for Surf Bay ($n = 11$), which is missing the September
2015 sample. Monthly composite $\delta^{18}O$ and $\delta D$ isotopes in precipitation throughout the year ranged from -12.3 ‰ to -4.8 ‰, and from -86 ‰ to -23 ‰, respectively. Monthly composite precipitation at each location was used to construct a local meteoric water line using $\delta^{18}O$ and $\delta D$ isotopes ($y = 7.571x + 5.527$; Fig. 3A) from monthly composite precipitation ($n = 47$). The range for winter $\delta^{18}O$ and $\delta D$ was from -8.6 ‰ to -6.6 ‰ and -61 ‰ to -40 ‰, respectively. Summer values of $\delta^{18}O$ and $\delta D$ in precipitation ranged from -12.3 ‰ to -5.3 ‰ and -86 ‰ to -38 ‰, respectively, and
fit within the range of historical isotopes in precipitation from the Falklands (GNIP; Fig. S5). Summer and winter $\delta^{18}O$ and $\delta D$ isotopes in precipitation ($n = 24$) passed tests for normality (Shapiro-Wilk, $p = 0.297$ and $p = 0.614$, respectively) and failed tests for equal variance (Fisher's F test, $p < 0.05$). A Mann-Whitney Rank Sum test indicated that the $\delta^{18}O$ isotopes in precipitation were not different for summer (median = -8.3 ‰) and winter (median = -7.4 ‰, $U = 39$, $p = 0.061$). For $\delta D$, the summer had a significantly lower median value (median = -64.3 ‰) than winter
(median = -46.5 ‰, $U = 22$, $p = 0.004$). A one-way ANOVA found no significant difference among sites in $\delta^{18}O$ (F



$_{(3,43)}$ = 0.323, p = 0.809) or δD isotopes (F $_{(3,43)}$ = 0.361, p = 0.785) in precipitation when samples from all months and sites were included (*n* = 47).

### 3.5 δ$^{13}$C and δ$^{18}$O of α-cellulose – temperature, humidity, precipitation

Across all sites, measurements of monthly average leaf oxygen and carbon stable isotope values for α-cellulose

extracted from leaf tissues (hereafter δ$^{18}$O$_{leaf}$ and δ$^{13}$C$_{leaf}$) had a strong positive correlation (Pearson's r = 0.877, p < 0.001, *n* = 24; Fig. 3B) and segregation between winter and summer values. Measurements of δ$^{18}$O in precipitation had no significant correlation with δ$^{18}$O$_{leaf}$ or δ$^{18}$O$_{root}$ across all sites (Table 1).

δ$^{18}$O$_{leaf}$ and δ$^{13}$C$_{leaf}$ values passed tests for normality (Shapiro-Wilk, p = 0.173 and p = 0.385, respectively) and equal variance (Fisher's F test, p = 0.865 and p = 0.196, respectively). Thus, a two-way analysis of variance was conducted

to detect the influence of independent variables (season and study location) on both δ$^{18}$O$_{leaf}$ and δ$^{13}$C$_{leaf}$. Season included two levels (summer and winter) and study location consisted of four levels (Bleaker Island, Cape Dolphin, Surf Bay, and West Point Island). Analysis of combined winter and summer δ$^{18}$O$_{leaf}$ had a mean of 28.9 ‰ ± 1.3 SD, and ranged from 26.3 ‰ to 31.8 ‰ (range of 5.4 ‰; Table S2). The effect of season was significant, with an F ratio of F $_{(1, 16)}$ = 183.2, p < 0.001, and a 2.6 ‰ difference between summer (mean = 30.1 ± 0.8 SD) and winter (mean =

27.5 ± 0.6 SD; Fig. 4; Table S2). The effect of study location yielded an F ratio of F $_{(3, 16)}$ = 4.8, p = 0.014, indicating a significant difference in δ$^{18}$O$_{leaf}$ among study locations. Pairwise multiple comparison (Tukey's post hoc test) of study locations indicated that Surf Bay is significantly more depleted in $^{18}$O$_{leaf}$ than Cape Dolphin (p = 0.016) and Bleaker Island (p = 0.029; Fig. 4). The interaction effect was not significant (p = 0.552). The mean of combined winter and summer δ$^{13}$C$_{leaf}$ value was -25.4 ‰ ±1.31 SD, ranging from -30.4 ‰ to -21.9 ‰ (range = 8.4 ‰; Table S2). For

δ$^{13}$C$_{leaf}$, there was a significant difference between seasonal values (F ratio of F $_{(1, 16)}$ = 40.8, p < 0.001) in summer (mean = -24.2 ‰ ± 1.05 SD) and winter (mean = -26.8 ‰ ± 1.3 SD, Fig. 4; Table S2). Study location (p = 0.861; Figs. 4B and 4D) and the interaction effect (p = 0.638) were not significant. The mean δ$^{13}$C in root α-cellulose (hereafter δ$^{13}$C$_{root}$; *n* = 14) for summer was -25.3 ‰ ± 1.27 SD, and -26.6 ‰ ± 1.38 SD in winter (Table S2). After δ$^{13}$C$_{root}$ data passed tests for normality (Shapiro-Wilk test; p = 0.085), but not equal variance (p < 0.05), the two-way ANOVA

indicated that for δ$^{13}$C$_{root}$, the effects for season (p = 0.201) and study location (p = 0.521) were not statistically significant. The interaction effect was not significant (p = 0.886).

The mean δ$^{18}$O in root α-cellulose (hereafter δ$^{18}$O$_{root}$; *n* = 14) for summer was 28.8 ‰ ± 1.04 SD, and 28.3 ‰ ± 0.5 SD for winter (Table S2). The δ$^{18}$O$_{root}$ data passed tests for normality (Shapiro-Wilk test; p = 0.483) and equal variance (Fisher's F test; p = 0.897); the two-way ANOVA indicated that for δ$^{18}$O$_{root}$, the location effect was statistically

significant, while season was not. The difference in mean values among seasons (F $_{(1, 8)}$ = 5.4, p = 0.049) and study location (F $_{(2, 8)}$ = 8.7, p = 0.010) were statistically significant. Pairwise multiple comparison (Tukey's post hoc test)



of study locations indicated that Cape Dolphin was significantly greater than Bleaker Island ($p = 0.012$) and West Point Island ($p = 0.049$). The interaction effect was not significant ($p = 0.397$).

## 4 Discussion

Significant inter-seasonal differences in $\delta^{13}C_{leaf}$ and $\delta^{18}O_{leaf}$ indicate that *P. flabellata* tissues record high-resolution patterns of environmental change, supporting the use of *P. flabellata* peat records as a paleoenvironmental proxy. The negative correlation between $\delta^{13}C_{leaf}$ and vapor pressure deficit suggests stomatal conductance is sensitive to atmospheric moisture conditions (Ferrio and Voltas, 2005). The observed positive correlation between $\delta^{13}C_{leaf}$ and temperature suggests higher temperatures led to an increased assimilation rate and reduced discrimination against $\delta^{13}C$

as shown in other vascular plant studies (Ferrio and Voltas, 2005; Ménot and Burns, 2001). $\delta^{13}C_{leaf}$ is driven by changes in the ratio of internal leaf partial pressure of $CO_2$ to that of ambient air, and can be explained by a greater influence of either stomatal conductance or increased photosynthetic capacity (Scheidegger et al., 2000). As plant stomata close in response to low humidity, the internal partial pressure of $CO_2$ decreases and the $\delta^{13}C_{leaf}$ increases (Farquhar et al., 1982, p.198).

The $\delta^{18}O_{leaf}$ is influenced by soil water, leaf water enrichment of $^{18}O$ from transpiration, and biochemical fractionations. Leaf water enrichment of $^{18}O$ due to transpiration, which is reflected in $\delta^{18}O_{leaf}$ (Deniro and Epstein, 1979; Roden and Ehleringer, 1999; Sternberg et al., 1986, p.198; Yakir, 1992), depends on relative humidity (Helliker and Ehleringer, 2002). However, the relationship between relative humidity and $\delta^{18}O_{leaf}$ deteriorated at relative humidity > 90 % in one C3 species (Helliker and Ehleringer, 2002). Diffusion limitation by stomatal resistance is

primarily driven by relative humidity (White et al., 1994). The $\delta^{18}O_{leaf}$ and relative humidity (> 60 %) were negatively correlated, which is consistent with other studies showing that $\delta^{18}O_{leaf}$ increases as relative humidity decreases (Barbraud et al., 2012; Helliker and Ehleringer, 2002). At high relative humidity the leaf will more strongly record variation in the isotopic composition of atmospheric vapor, however we have no direct measurements of $\delta^{18}O$ in water vapor.

Previous work on leaf water and cellulose isotopes in grasses demonstrated that the atmospheric-leaf vapor conditions are a strong predictor of $\delta^{18}O_{leaf}$ (Helliker and Ehleringer, 2002; Lehmann et al., 2018). The $\delta^{18}O$ of the leaf water is captured in the cellulose isotopes and can reflect the effect of changing environmental conditions during the growth of the leaf (Helliker and Ehleringer, 2002; Lehmann et al., 2017). The $\delta^{18}O_{leaf}$ can also depend on physiological effects, the type of plant anatomical feature used, and stage of development (Lehmann et al., 2017; Liu et al., 2017). Because

cellulose records environmental variation along a gradient during leaf growth, we collected and homogenized whole leaves to avoid the complications of $\delta^{18}O$ enrichment (Helliker and Ehleringer, 2002). Our work supports the finding that atmospheric-leaf vapor conditions are reflected in $\delta^{18}O_{leaf}$, and expands the use of such paleoclimate proxies to peat-forming tussock grasses, which opens up new possibilities for reconstructing paleoclimates across the South



Atlantic and beyond. At the higher relative humidity range, the $\delta^{18}O_{leaf}$ is more of a reflection of source water, while
the $\delta^{18}O_{leaf}$ at low humidity differed greatly from source waters because of evaporative enrichment of $^{18}O$.

The positive correlation between $\delta^{13}C_{leaf}$ and $\delta^{18}O_{leaf}$ (Pearson's r = 0.88, p < 0.001, $n$ = 24; Table 1; Fig. 3B) suggests
that stomatal conductance is the driving force acting on these two proxies, which is a likely scenario when water is
not limiting (Saurer et al., 1997; Scheidegger et al., 2000). According to the Scheidegger et al.(2000) model, the
decline in stomatal conductance was much more strongly expressed than photosynthetic capacity (maximum net
photosynthesis). When air humidity increases, stomatal conductance is assumed to increase. In our study, stomatal
conductance is likely driving both $\delta^{13}C_{leaf}$ and $\delta^{18}O_{leaf}$ due to relative humidity (Fig. 5). This pattern fits well with the
Barbour and Farquhar (2000) model. Measurements of relative humidity allowed us to determine that stomatal
conductance was more influential as a possible cause of change in partial pressure of $CO_2$ within the leaf.

In the Falklands, precipitation amount is not highly seasonal, but tends to vary the most in the summer (Fig. S6; data
from Jones et al., 2013). Over the year of our study, summer $\delta^{18}O$ in precipitation varied more than winter $\delta^{18}O$ and
tended to be more depleted. This is supported by the wind rose from Bleaker Island, which indicated prevailing winds
from the SW (Fig. S2). In winter, precipitation tended to be more enriched in $^{18}O$, and most prevailing winds came
from the west and NW where $^{18}O$ in equatorward precipitation would be more enriched than $^{18}O$ in high latitude
meteoric sources (Fig. S5). The significant negative correlation between $\delta^{18}O$ in precipitation and $\delta^{18}O_{root}$ at Cape
Dolphin (Pearson's r = 0.868, $p$ = 0.025, $n$ = 6; Table 1) is consistent with the latitudinal origin of storm tracks and
seasonal wind data (Fig. S4; Fig. S2). The pattern found at Bleaker Island is less clear, and warrants further
investigation.

The observed lack of correlations between $\delta^{18}O_{leaf}$ and $\delta^{18}O$ values of precipitation (Table 1) demonstrate the
overriding influence of humidity on patterns of leaf water $\delta^{18}O$. An alternative explanation for a lack of correlation is
that our precipitation sampling density was not sufficient to establish a relationship between $\delta^{18}O_{leaf}$ and precipitation
$\delta^{18}O$. Although *P. flabellata* produces new leaves throughout the year, the growth rate of leaves may not be in sync
with shorter precipitation sampling intervals (less than monthly composite precipitation). Examination of leaf waters
post-precipitation events would improve our understanding of $\delta^{18}O_{leaf}$.

The observed relationship between $\delta^{18}O$ in precipitation and $\delta^{18}O_{root}$ appeared to be less clear in part due to low sample
number from only three study locations (Bleaker Island, Cape Dolphin, and West Point Island). At Bleaker Island,
there was no correlation between $\delta^{18}O$ in precipitation and $\delta^{18}O_{root}$, while Cape Dolphin had a strong negative
correlation (Table 1). At Cape Dolphin, greatest enrichment of $^{18}O_{root}$ occurred in summer when $\delta^{18}O$ precipitation
was relatively low. In contrast, at Cape Dolphin $\delta^{18}O_{root}$ was lowest during the winter months when $\delta^{18}O$ of
precipitation was high. Despite the strong relationship between $\delta^{18}O$ precipitation and $\delta^{18}O_{root}$ at Cape Dolphin, we
propose that *P. flabellata* 'bogs' may not be ombrotrophic, and may potentially source water from fog, sea-spray,
groundwater, or a mix. However, the relationship found at Cape Dolphin warrants further analysis of $\delta^{18}O_{root}$ and
source waters. We also consider that the $\delta^{18}O_{root}$ records a signal of leaf humidity, and is influenced by source water
and humidity effects on leaves; sugars produced in leaves are transported to roots to form cellulose, and about half of



the oxygen atoms in root cellulose originate from the leaf water signal. Roots of *P. flabellata* may have represented
greater temporal integration of $\delta^{18}O$ from precipitation into cellulose due to difficulty in distinguishing new growth in
roots, like *Empodisma* in New Zealand (Amesbury et al., 2015).

Further work is needed to understand the relationship between $\delta^{18}O_{root}$, $\delta^{18}O$ of precipitation, and that of root and leaf
waters. Identifying sources of potential water would also add value, especially considering anecdotes of local
differences in fog in the Falklands, which may be an unappreciated source of water for *P. flabellata*. Across the
geographic range in the South Atlantic, *P. flabellata* may record a larger latitudinal gradients of isotopes in
precipitation, as well as temperature and humidity, than recorded in the Falklands. Thus, establishing the seasonal
patterns recorded by *P. flabellata* cellulose in the Falklands enables critical testing of paleoclimate hypotheses
regarding the dynamics of Southern Hemisphere westerly wind behavior from broader latitudinal and longitudinal
locations where *P. flabellata* occurs. Although *P. flabellata* forms high resolution peat records and is sensitive to inter-
seasonal differences, reconstructions using *P. flabellata* peat would represent an integrated signal of broader climate
trends, and not inter-annual or lower frequency differences.

## 5 Conclusion

The scarcity of terrestrial paleoclimate records in the South Atlantic has limited our understanding of past and future
climate change and its impacts on ecosystems and people. We found that carbon and oxygen stable isotope values in
*P. flabellata* tissues are correlated with seasonal differences in temperature and moisture, providing a promising new
avenue for paleoclimate reconstructions in the South Atlantic. *P. flabellata* peats have high accumulation rates, contain
abundant leaves, and date back to at least 12,500 [14]C years, with the potential to provide decadal-scale records of
temperature, precipitation, and moisture source. Future work is needed to determine whether $\delta^{18}O$ and $\delta^{13}C$ of cellulose
from *P. flabellata* macrofossils complement other regional proxies for changes in atmospheric temperature and
relative humidity during the Holocene. Troublingly, these coastal peatlands are currently threatened by sea level rise
and over-grazing, and their reductions means we are losing vital information about past environments in a time when
paleoclimate records are needed to provide context for modern climate change in the South Atlantic.

## 6 Data availability

Datasets for monthly stable isotopes in precipitation, average temperature, and average relative humidity have been
submitted to the Global Network of Isotopes in Precipitation (https://nucleus.iaea.org/wiser) and will be publicly



available upon acceptance for publication. Datasets for leaf and root stable isotopes of cellulose can be found at http://dx.doi.org/10.5281/zenodo.3104573, hosted at Zenodo upon acceptance for publication.

## 7 Author contribution

DG, DW, and JG designed the experiments and DG carried them out. DG performed laboratory analyses. DG prepared
the manuscript with contributions from all co-authors.

## 8 Competing interests

The authors declare that they have no conflict of interest.

## 9 Acknowledgements

We gratefully thank the citizen scientists who collected samples: Ben Bernsten, Nikki and Mike Summers, Mike and
Phyl Rendell, Robert and Elaine Short, Kicki Ericson, Thies Matzen. Paul Mayewski at the University of Maine
Climate Change Institute provided precipitation collectors. Logistical, field, and laboratory support was provided by
Paul Brickle and Megan Tierney at the South Atlantic Environmental Research Institute, Craig Cook at the University
of Wyoming Stable Isotope Facility, Kayla Greenawalt, and Jiemin Guo. Funding: This research was supported by
the US National Science Foundation [grant numbers DGE-1144423 and EF-1137336].

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



**Table 1.** Correlation coefficients (Pearson's r) of $\delta^{18}O$ and $\delta^{13}C$ in leaf and root cellulose between $\delta^{18}O$ in monthly composite precipitation, monthly average temperature, and humidity by site. Bold values indicate significant correlations >0.600 Pearson's r. Significance level is $p < 0.05$.

| | Site | $n$ | $\delta^{13}C_{leaf}$ | $\delta^{18}O_{leaf}$ | $\delta^{18}O$ precipitation | Temp | Relative Humidity |
|---|---|---|---|---|---|---|---|
| $\delta^{18}O_{leaf}$ | All sites | 24 | **0.877 (<0.001)** | -- | -0.201 (0.346) | **0.889 (<0.001)** | **-0.877 (<0.001)** |
| | Bleaker Is. | 6 | **0.864 (0.026)** | -- | -0.058 (0.913) | **0.947 (0.004)** | **-0.939 (0.005)** |
| | Cape Dolphin | 6 | **0.990 (<0.001)** | -- | -0.357 (0.487) | **0.877 (0.021)** | **-0.979 (<0.001)** |
| | Surf Bay | 6 | 0.769 (0.074) | -- | -0.378 (0.460) | **0.952 (0.003)** | **-0.977 (<0.001)** |
| | West Point Is. | 6 | **0.971 (0.001)** | -- | -0.330 (0.523) | **0.977 (<0.001)** | **-0.900 (0.014)** |
| $\delta^{18}O_{root}$ | All sites | 14 | -- | 0.385 (0.174) | -0.302 (0.294) | 0.311 (0.279) | -0.217 (0.457) |
| | Bleaker Is. | 6 | -- | 0.243 (0.642) | 0.222 (0.672) | 0.219 (0.677) | -0.116 (0.827) |
| | Cape Dolphin | 6 | -- | 0.623 (0.186) | **-0.868 (0.025)** | 0.701 (0.120) | -0.694 (0.126) |
| | Surf Bay | -- | -- | -- | -- | -- | -- |
| | West Point Is. | 2 | -- | -- | -- | -- | -- |
| $\delta^{13}C_{leaf}$ | All sites | 24 | -- | **0.877 (<0.001)** | -- | **0.817 (<0.001)** | **-0.759 (<0.001)** |
| | Bleaker Is. | 6 | -- | **0.864 (0.026)** | -- | **0.843 (0.035)** | -0.688 (0.131) |
| | Cape Dolphin | 6 | -- | **0.990 (<0.001)** | -- | **0.849 (0.032)** | **-0.952 (0.003)** |
| | Surf Bay | 6 | -- | 0.769 (0.074) | -- | 0.780 (0.067) | **-0.819 (0.046)** |
| | West Point Is. | 6 | -- | **0.971 (0.001)** | -- | **0.977 (<0.001)** | **-0.816 (0.047)** |
| $\delta^{13}C_{root}$ | All sites | 14 | **0.724 (0.003)** | -- | -- | 0.492 (0.074) | -0.299 (0.300) |
| | Bleaker Is. | 6 | **0.832 (0.039)** | -- | -- | 0.561 (0.247) | -0.273 (0.601) |
| | Cape Dolphin | 6 | 0.570 (0.237) | -- | -- | 0.778 (0.068) | -0.718 (0.108) |
| | Surf Bay | -- | -- | -- | -- | -- | -- |
| | West Point Is. | 2 | -- | -- | -- | -- | -- |






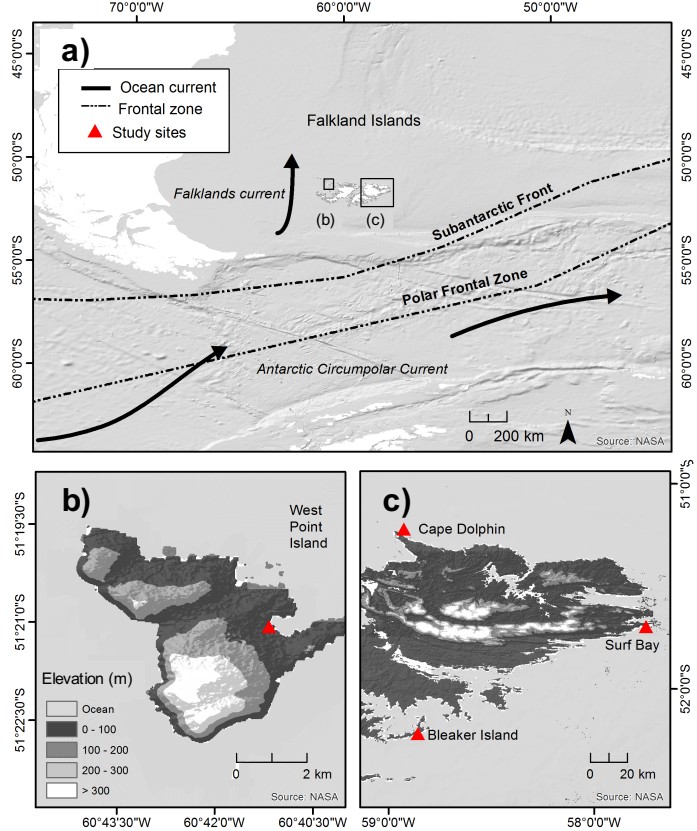

**Figure 1.** Study region. **a**) Map of the Falkland Islands and western South Atlantic Ocean with ocean currents (black arrows) and frontal zones (dashed lines). Study sites are shown in: **b**) West Point Island and **c**) Bleaker Island, Cape Dolphin, and Surf Bay.



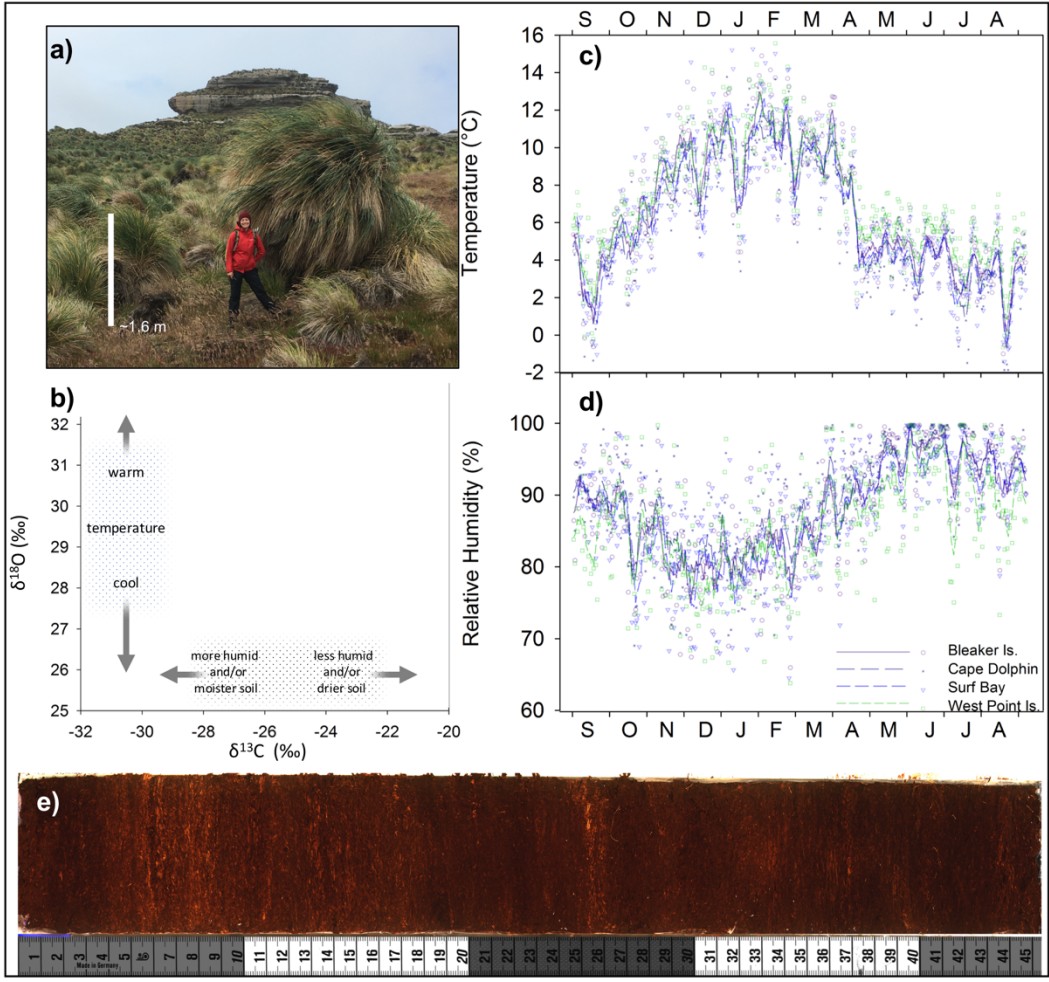

**Figure 2. a)** A single large *Poa flabellata* bog made up of dead and living grass tillers growing on top of a decomposing pedestal at Cape Meredith, Falkland Islands. **b**) Conceptual figure of the relationship between $\delta^{13}C$ and $\delta^{18}O$ of leaf cellulose and air temperature, and humidity and/or soil moisture. **c)** Daily average temperature (ºC) and **d)** relative humidity (%) from September 2015-August 2016 at the four study locations calculated from 2 hour measurements. Seven day running averages of daily average temperature and relative humidity are indicated by the lines for each study location. **e)** Image of a *P. flabellata* peat core section (Photo: D. Groff).



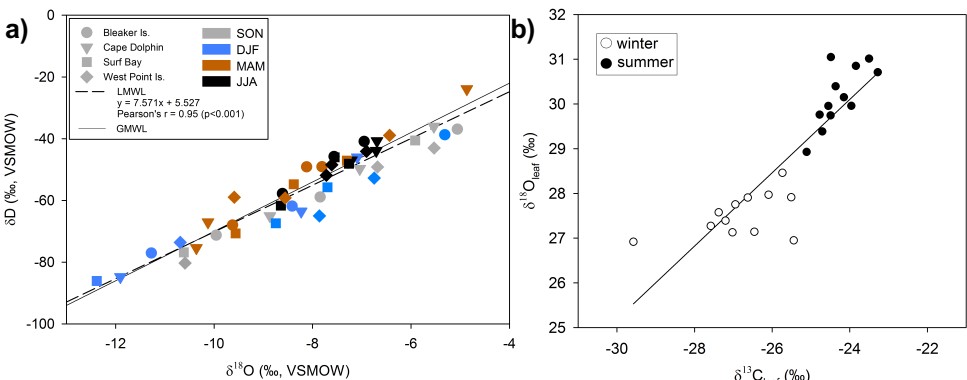

**Figure 3. a)** $\delta^{18}O$ and $\delta D$ (‰, VSMOW) isotopes in precipitation for each location (symbol shape) during four seasons (symbol color). The constructed local meteoric water line (LMWL; $y = 7.571x + 5.527$) is shown as a dashed line and global meteoric water line (GMWL: $\delta D = 8.0\ \delta^{18}O + 10$) is a solid line. **b)** The relationship between average $\delta^{13}C_{leaf}$ and $\delta^{18}O_{leaf}$ (Pearson's correlation coefficient, $r = 0.877$, $p < 0.001$, $n = 24$). Open circles are average values for samples collected in winter, solid circles in summer.

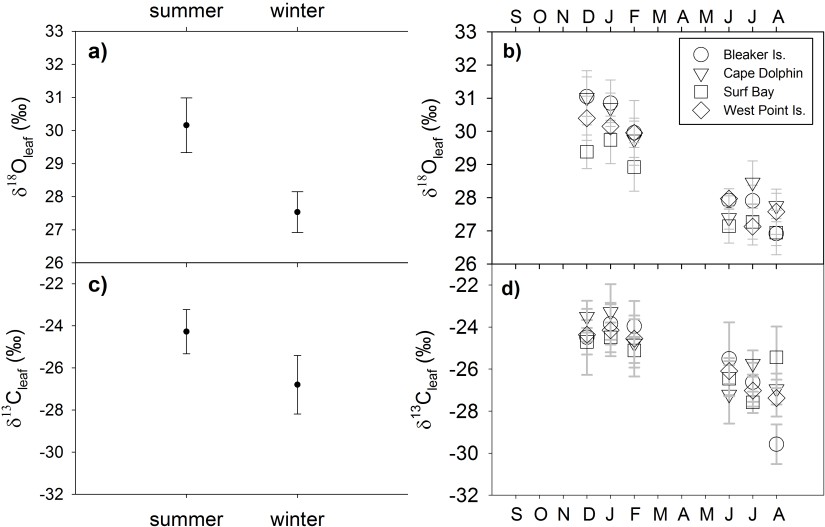

90    **Figure 4.** Oxygen and carbon stable isotopes of *Poa flabellata*. **a)** $\delta^{18}O_{leaf}$ (‰) comparison (mean ± 1 SD) between summer (DJF) and winter (JJA) and, **b)** at four study sites over one year; **c)** $\delta^{13}C_{leaf}$ (‰) comparison (mean ± 1 SD) between summer and winter, and **d)** at four study locations over one year.

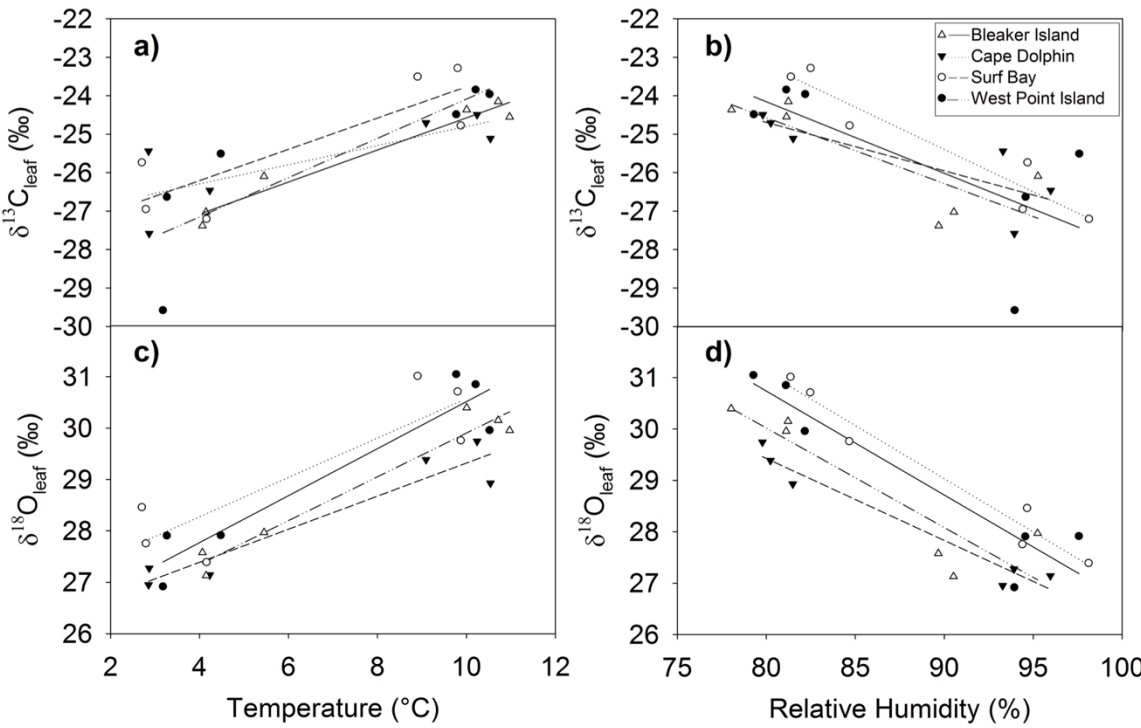

95 **Figure 5.** Relationship between carbon and oxygen stable isotopes in cellulose and temperature and relative humidity during winter and summer at four sites. Relationship between $\delta^{13}C_{leaf}$, **a)** temperature and **b)** humidity; Relationship between $\delta^{18}O_{leaf}$, **c)** temperature, and **d)** humidity. Use Table 1 for Pearson's r and p-values corresponding to correlations for each site.