# Peer review of "Modern calibration of *Poa flabellata* (Tussac grass) as a new paleoclimate proxy in the South Atlantic"

_Biogeosciences, 2019_

## Referee Comment (RC1) · Anonymous Referee #1 · 28 Feb 2020

Overall, this is a really nice dataset that has a lot of promise for enhancing our understanding of paleoclimate in the southern hemisphere. I have several minor comments and a few major comments below. There is a lot of analysis into explaining the variation in the isotopes, how that's controlled by plant physiology, but not much discussion and explanation of how these isotope signals will be used to reconstruct paleoclimate, especially in context of applying this to a peat core (through time). Providing a roadmap for how changes in d13C and d18O will be interpreted would be useful and weather this is qualitative or can it be pushed further to be quantitative? There is the suggestion that this is going to really help us understand climate dynamics, but then there is not discussion of how. Is this going to provide temperature or relative humidity or both

(there is not clear indication of which and both are correlated with the isotopes) and how to you disentangle any changes in source water d18O through time?

Some discussion of how to do this for paleoclimate also needs to focus on how this study shows nicely that the leaves are recording a seasonal signal. So, when you go down core, how are you going to deal with this? Are you going to focus on a large sampling of leaves from each horizon (age?) with the expectation that you are sampling both seasons or is it going to be a single multiple leaf measurement to approximate an annual signal? Some thought into this is needed as the data analysis and presentation may need to be added to or adapted for paleo work. I'd like to see a clearer connection between this nice modern calibration data and how to use it for the past.

Maybe these comments don't seem fair (the focus is rally on the modern calibration), but the title, abstract, intro, and conclusion suggest this is a new great paleoclimate proxy. My guess is the authors will be applying this to a peat core in the future. It would be nice if the framework is put forth here. If that can't be done, then I think the paleoclimate proxy significance and mention needs to be removed throughout.

Line 28: "trends in southern hemisphere climate dynamics" – is that consistent with what you can actually do with this proxy? Or is it something more specific?

Lines 43-46: Awkward sentence with semicolon connecting two separate statements.

Line 56: Is it really called a "bog"? That's not confusing... It's hard to reconcile this description with the one line 70 and "pedestal" which is in the caption for Figure 2. Maybe some annotation on the figure or more description would be useful. I'd like to have a clear idea how this is going to develop over time in a peatland and how this plants growth habit is going to translate into a vertical succession (or some crazy patchwork of different ages in a peat core).

Line 57: Something wrong with new sentence that starts here and sentence seems incomplete too.

Line 56-57: Either they use precip or the precip wets all that organic matter and then there is evaporative enrichment b/c it is exposed to wind/sun.

Line 70: Maybe start a new paragraph here or have a better transition?

Line 71 and below: check the order in which isotopes are first described. Here delta symbols are used first but aren't defined, next sentence doesn't use delta symbols (carbon isotopes), and then defined on line 90-91. I think this comes up a few other places and would be worth cleaning up.

Lines 92-93: Improving "westerly wind dynamics" is different than what's mentioned elsewhere. What is it that this new proxy can solve and make it consistent throughout.

Line 100: Could the km hr-1 also be reported here and later for reference? Not to many readers will think about wind speed in m/s.

Line 170-172: How are temperature and humidity related? Based on the figure, they look highly correlated. If they are, then how do you disentangle their effects from the cellulose d13C and d18O as they are both strongly related? I didn't see any multiple regression analysis reported below either.

Line 186: is west, NW, and SW 79%? That's missing from the sentence. Reporting 21% for the last source and not saying anything about the other 3 directions is reads strangely and compared to the prior sentence.

Line 206-207: I think you need to be really careful presenting this here and then in the discussion below. With this data, maybe the other factors have a stronger control than precipitation d18O, but at least at some level, precip d18O must be important. So, when applying this down core (through Holocene), if there are changes in d18O, they must change the cellulose d18O (and then it's probably modified by the other factors you report here). I think this is critical to point out for those who will use this in paleo applications. More on this below.

Section 3.4: there's no mention here of the relationships between the isotopes and

temp and humidity but these are in the discussion, figures and tables. This would be a good place to describe the relationships of to both environmental factors.

Line 237: What negative correlation? Not in the results or the figures. VPD is not discussed prior to this.

Line 242-244: Is this consistent with the "low" humidity of the Falklands of >70%?

Lines 283-288: Relating plant tissue d18O (or dD) to precipitation is always a challenge. Even if you had leaf water or soil (pedestal?) water, it would still be complicated, but maybe give some insight. Many studies try to relate d18O of the plant back to precipitation, but here, it's clear that other factors modify this. But, at the most basic level, d18O precip is setting source water and then maybe there is mixing with other sources (ground water, dew, etc), but that is then modified by temp/humidity, etc. I think some discussion here is needed to highlight that this is much more complicated than indicated for the reader. If one tries to do this down core, changes in d18Oprecip must at some level matter for the d18O of plant source water and ultimately the d18O cellulose. Also, getting into event precip (as mentioned) could be interesting, but it might be more informative to pull into this discussion when the leaves/cellulose are being made. Can you say anything about this with the data in hand?

Overall, the discussion is lacking a clear description of how the d13C and d18O would be used to interpret paleoclimate. Is it a temperature signal, a humidity signal, a source of precipitation signal? Or is it all of the above? How will a down core record be interpreted? Is there any way to put some uncertainty into this? How are you going to disentangle the multiple correlations between the isotopes themselves and the relationships with temp and humidity?

Figure 1: It would be nice here or elsewhere to have the wind diagrams and the precip source isotopes provided. I don't know what the figure limitations are for this journal, so maybe that's not possible. But, it sure would be nice to have a bit more of the great data collected here summarized in the main article figures.

Figure 2: It would be nice if the interpretive strategy figure here was where that data is reported. The peat core is interesting, but not really discussed. It would be nice if it was to put into an interpretive strategy that could be used for downcore paleo reconstructions.

Figure 3: VSMOW on 3a, but VSMOW and VPDB missing on 3b. For the LMWL reported here, can you report the r or R2, p-value, and n?

Figure 4, VSMOW and VPDB needed

---

## Referee Comment (RC2) · Samuel Bodé (Referee) · 23 Mar 2020

I believe this is an interesting piece of work, as indeed more reliable paleoclimatic proxies for the southern Atlantic are needed, to increase our understanding of past climate patterns. The author also collected a nice dataset. I do agree that Poa flabellate peat has promising potential to be the base of a good proxy, is it has a high accumulation rate in the peat, and is mainly present as the unique plant species. I do however not agree that the real poof of the power of the recorded isotopic signal as paleoclimate proxy has really been given in this manuscript. I have a couple of major concerns on the data treatment and interpretation and a large number of minor remarks. First, the

observed correlation of 13C and 18O of the leaf cellulose with RH and T is used as an indication of the power of the proxy for paleo climatic studies. The leaf samples were young leafs growing during the sampled year. The leaves start to grow vertically in the summer and get broader during the winter. The summer samples are thus systematically younger samples than the winter samples. It can not be excluded that the observed difference in 18O and 13C is related to the change in leaf phenology rather than a climatic response. As the entire leaves are collected in winter, the recorded isotopic signal is a combination of the entire growing season. Further, it is important to note that this seasonal resolution will not be recorded in the peat record, as only mature leaves will contribute to the litter. A much better way of assessing the potential of the proxy for paloclimatic reconstructions would of course be to sample a peat core, and correlate 13C and 18O signals of the core to recorded climate data. Therefore, it is needed to better frame the study, and rather use it as a background study on the physiological response of the Tussac grass and incorporation of atmospheric isotopic signal in the cellulose and only put it forward as a very first step toward the development of a paleo climatic proxy. Further comments: It is not always clear what is tested when statistical tests are performed. An example of this, is when the RH and T of the different locations are compared. It seems to me that the yearly average T and RH are compared using the individual days as replicates (I.e. SD computed on variation between days). This seems wrong to me as the variation in T and RH between individual days has no link with the uncertainty on the average T and RH of that location. To compare the RH and T of the measuring period for these locations only the measurement uncertainty (which is typically very small). Further to be able to say something on the difference in yearly average RH and T between sites in general, several years of observations are needed. The same problem occur when 18O and 2H in precipitation between seasons is compared, the monthly variability is not related to the uncertainty of the mean. On top of this, it is not so meaningful to compare the numerical average when comparing seasons, i.e. the weighted average should be used. Again only the measurement uncertainty is relevant when comparing the seasons. Using the weighed

averages the average of the locations could be used to compare the different seasons, however, to me it doesn't seems right to use these different location as replicates. (as a final note on this seasons start and end the 21th of a month, while samples were taken per month, this should also be acknowledged.) When looking at the correlation of the 13C and 18O in leaf cellulose with RH and T, it is observed that they correlate with both. Beside the issue discussed above it is also important to note that RH and T are also strongly correlated (i.e. Drier (77 -85 %)/hotter (9-11°C) summers and wetter (90-98%)/colder (1-5.5°C) winters). For which, from the data it can not be concluded if the effect is a result of the RH or T or both. It would be interesting to give some insight on what effect might be prevailing here.

Smaller remarks: L75: it sounds quite contradictory to expect low WUE in water-limited environments. L80: It is a rather strange thing to say that $\delta$18O of source water often correlate with temperature. Better to say (and this is also how it is described in the given reference) that the $\delta$18O of precipitation correlates with temperature. (sure in the case of tussocks, source water is directly related to precipitation, but this cannot be claimed in general). L128: It is not clear what 'frozen for eight days' exactly means, what happened after these 8 days? In fact the entire section is not very clear, what is the point of freezing them for 8 days if some where already stored at RT for 6 months? Please rewrite and clarify. L144 and L150: When secondary reference material are used, the accepted value used for it should be given (this do sometimes change over time). L203: Paragraph is too long, to many irrelevant details are given for which the major lines get lost. I believe this paragraph should be shortened by c.a. 50% L204: Why is this correlation analysis done on averaged values? It should be done using the individual data points.

L13: replace 'investigate' by 'measured', delete 'plants' L14: I believe the author mean: '…... explore relationships with seasonal temperature and air humidity variations across 4….' L16: Delete 'significantly' (if not you would not report it) L23: 'did not differ significantly' (there is no test to claim that 2 things do not differ, you can only claim

that you could not see a significant difference. L32: '...resulted in an intensification and polward....' L35: Sentence starting with 'The inconsistency of....' Is not totally clear, reformulate. L47: Should it not be '....generate substantial amounts of peat...' ? L57: I beleve it sould be cited as. 'Smith and Prince (1985) established radiocarbon...' L62: '....of any peatland, globally, P......" L67: '....in this semi-arid habitat...' L69: a) I guess it is 'up to 39 cm' or 'c.a. 39 cm' b) '...year) while in winter an increase....' L70: '....tiller at the base of....' L70: Sentence starting with 'The climate signal....' Is not clear, please reformulate. L73: 'physiological responses such as changes in stomatal conductance and....' L77: '...information on...' L78: '...tissue was formed...' L79: '...humidity and plant physiology (...' L80: '...often correlate...' L81: '...cellulose can also...' L82: '....plant water pools.' L86: rather use '‰L97: If mean temperature is given, the time span of this mean should be given, same for L109: Why is the fact that they were first shipped to University of Maine mentioned? I don't think the reader is interested In the postal rout.... L113: It is '...were purified...' or 'Water was extracted out of precipitation sample....' L119: '...relative to VSMOW.' No need to mention they are reported in ‰ (this is visible in the results, and can lead to inconsistency). L130: '...were used, fine roots...' L133: Just out of personal curiosity, did the grinding of leaf material using a ball mill work? my experience is that this do not work very well with fibrous material. L137: a) What is an undetectable amount? Should give a detection limit here, if not it is meaningless. b) it is nitrogen and carbon content, not %nitrogen an %carbon L140: 'varied by < 0.1 ‰ and 0.3 ‰ respectively' L143: '...relative to VSMOW (....' L146 and 151: replace ranges by 'c.a. 0.25 mg' and 'c.a. 2 mg' L148: Delete '; units are expressed..........mil)' L150: Reformulate last sentence L153: what is meant with 'analysing the average'? I think the author means that for every site 3 to 4 leafs were used as replicates in every month. L184: what is the 'n = 344', summer only counts 90 days.... L189: Simply say that September 2015 sample was missing for surf bay (we all know that a year has 12 months). Was that sample missing or was it not sampled, meaning that October 2015 is in fact September + October? L192: '....$\delta$D values (...' , 'monthly composite' is redundant with the first part of the

sentence. L204: '…..leaf $\delta$18O and $\delta$13C values…' L206: add p value after segregation between winter and summer values. L206: How could measurement have a significant correlation? I believe the author meant to write that the d18O value of precipitation did not correlate significantly. Further it would be more logical to write that d18O of leaf and root did not correlate with d18O of precipitation rather than the other way around (statistically it is the same thing, though it is not logical). L212: '28.9 $\pm$ 1.3 ‰ídem at L214 etc….(change everywhere) L213: I do not find it meaning full to add the range, as the distribution was normal, giving average $\pm$ SD is enough (and really no reason to add also the range). L222: interaction is not an effect, what the author wanted to say is that 'no significant interaction could be observed'. L237: Not clear if this is an own observation (nothing is mentioned about VPD in the result section) or something from literature. L241: '…ratio of CO2 partial pressure in the leaf and that of the ambient air….' L241: what can be explained, the difference in parial pressure ratio or the effect of it on the 13C? L257: what are 'cellulose isotopes'? L317: Quite strange to say 'at least 12,500 14C years, while on line 64 it says that peatlands initiated between $\sim$ 12,500 and 5,500 14C years…. Figure 3b: Why is not the individual data presented, rather than averages? Table S1: if the it is given that longitude is south, the negative sign should not be used (sensu stricto a negative south latitiude is a northen latitude). So remove the '(S)' or the '-' idem for long (W). Figure S3: Link did not work, until I found that the '-' was not for a split for a line break (like my browser interpreted it when clicking on it), but a real hyphen. Probably better to use 'https://climatereanalyzer.org/' Figure S5: Add your data to this graph.

---

## Author Comment (AC1) · 13 Apr 2020

Dear Anonymous Reviewer,

We are grateful for your helpful comments, which have improved this manuscript. We have responded to your comments below. We also provide an explanation of the changes we intend to make in the manuscript.

Best wishes, Dulcinea Groff

\* There is a lot of analysis into explaining the variation in the isotopes, how that's controlled by plant physiology, but not much discussion and explanation of how these

isotope signals will be used to reconstruct paleoclimate especially in context of applying this to a peat core (through time). Providing a roadmap for how changes in d13C and d18O will be interpreted would be useful and weather this is qualitative or can it be pushed further to be quantitative?*

Peat-based reconstructions may be limited to identifying periods of warm/dry or cold/wet conditions that are more extreme than our observed seasonal variations (or more similar to them). For now, this proxy remains qualitative, but more work could be done to evaluate this proxy to assess its suitability for quantitative reconstructions (perhaps with leaf wax or alkenone biomarkers, though our preliminary data on hydrogen isotopes in precipitation suggests this may not be feasible). Resolving temperature and moisture signals independently would likely require growth chamber experiments.

Solution: We agree that an expanded discussion on paleoclimate reconstructions would be helpful. We intend to add a flow-chart and/or conceptual figure that outlines exactly what we think this proxy could reconstruct, including time averaging, and what the methods should look like i.e. bulk, individual leaves, different processing needed.

*There is the suggestion that this is going to really help us understand climate dynamics, but then there is not discussion of how. Is this going to provide temperature or relative humidity or both (there is not clear indication of which and both are correlated with the isotopes) and how to you disentangle any changes in source water d18O through time?*

Thank you for pointing out that this is unclear in the manuscript. The oxygen and carbon isotopic signatures are positively correlated in our study, which represents a modern snapshot of conditions Fig. 2B). Based on this modern calibration work, we recommend measuring C and O values together, and that these values would identify the timing of transitions between warmer and drier to colder and wetter conditions that correlate with major hemispheric drivers in climate.

Solution: We intend to add a section to the Discussion that more clearly outlines how

this proxy would work in practice, including the conceptual figure/flowchart mentioned above.

We didn't find a relationship between d18O in precipitation and the d18O of leaf cellulose in our 1-year study. It is possible that because the d18O in leaf water is controlled by source water and humidity, any changes in humidity confound a direct relationship between source water d18O values and leaf water/cellulose d18O. It may also be possible that the variation in source area did not affect the d18O enough for us to detect a significant impact on the d18O of the leaf cellulose. Based on this work, the source area could influence leaf cellulose d18O value in peat records if source area changes were greater than what we observed. That's true of all oxygen isotope reconstructions, though, and is not a unique consideration for the Southern Ocean.

Solution: We intend to make this clearer in the text.

* Some discussion of how to do this for paleoclimate also needs to focus on how this study shows nicely that the leaves are recording a seasonal signal. So, when you go down core, how are you going to deal with this? Are you going to focus on a large sampling of leaves from each horizon (age?) with the expectation that you are sampling both seasons or is it going to be a single multiple leaf measurement to approximate an annual signal? Some thought into this is needed as the data analysis and presentation may need to be added to or adapted for paleo work. I'd like to see a clearer connection between this nice modern calibration data and how to use it for the past.*

We recommend incorporating a large sampling of leaves from a horizon (1 cm), and expect that to be a sampling of both seasons, incorporating several years. These peat records are highly productive, but even so, time averaging within a given 1-cm level should be greater than than the age of a leaf. In the peat records we've analyzed, we've commonly found sedimentation rates to be ∼20-30 years/cm, with most records extending at least 13,000-15,000 years old. As with any prehistoric reconstruction, it is important to consider temporal grain and resolution, which is going to constrain the

inferences you can make.

Solution: We intend to include a section with a recommended workflow and considerations for applying to peat records, including our recommendations for methods (e.g., including that multiple whole leaf fragments should be used from each level, and methods for cellulose extraction and purification).

Line 28: "trends in southern hemisphere climate dynamics" – is that consistent with what you can actually do with this proxy? Or is it something more specific?

Based on what we have established with this study, this proxy can indicate trends in conditions similar to what we observed seasonally: warm/dry, cool/wet, which is more specific than our generalized statement. Multiple paleo-records could point to changes in Southern Hemisphere climate dynamics, but can not necessarily resolve the drivers of those dynamics.

Lines 43-46: Awkward sentence with semicolon connecting two separate statements.

Thanks for pointing this out. Solution: We will edit for clarity.

Line 56: Is it really called a "bog"? That's not confusing... It's hard to reconcile this description with the one line 70 and "pedestal" which is in the caption for Figure 2. Maybe some annotation on the figure or more description would be useful. I'd like to have a clear idea how this is going to develop over time in a peatland and how this plants growth habit is going to translate into a vertical succession (or some crazy patchwork of different ages in a peat core).

Yes, colloquially each pedestal is called a "bog," and we will remove this from the manuscript to avoid confusion – especially because the tussac peat that forms the soils in these stands is not a bog, either. We will call it a "pedestal" throughout. ("Tussac/tussock" is already confusing enough.)

The taphonomy of these pedestals is poorly understood. We intend to incorporate this in our conceptual figure of tussac pedestals and peat. There could be a patchwork of

ages within a peatland, but as of yet we have not found any evidence of age reversals or other chronology problems in our cores.

Line 57: Something wrong with new sentence that starts here and sentence seems incomplete too.

Thank you for pointing this out. Solution: This should read: Smith and Prince (1985) previously established radiocarbon (14C) dates for a P. flabellata pedestal and estimated an age of 250 to 330 years.

Line 56-57: Either they use precip or the precip wets all that organic matter and then there is evaporative enrichment b/c it is exposed to wind/sun.

Fair point. Solution: We will edit the text to indicate this.

Line 70: Maybe start a new paragraph here or have a better transition?

Will do, thanks.

Line 71 and below: check the order in which isotopes are first described. Here delta symbols are used first but aren't defined, next sentence doesn't use delta symbols (carbon isotopes), and then defined on line 90-91. I think this comes up a few other places and would be worth cleaning up.

Thanks for noting this. We will make this fix for consistency.

Lines 92-93: Improving "westerly wind dynamics" is different than what's mentioned elsewhere. What is it that this new proxy can solve and make it consistent throughout.

Great point. Solution: We intend to make the text clearer about exactly what climate conditions and questions this proxy can inform, while still placing it within the broader discussion of Southern Hemisphere climate dynamics.

Line 100: Could the km hr-1 also be reported here and later for reference? Not to many readers will think about wind speed in m/s.

The International System of Units recommends m/s, so we will retain use of this standard for consistency with them, and with other studies. (https://physics.nist.gov/cuu/Units/units.html)

Line 170-172: How are temperature and humidity related? Based on the figure, they look highly correlated. If they are, then how do you disentangle their effects from the cellulose d13C and d18O as they are both strongly related? I didn't see any multiple regression analysis reported below either.

As discussed above, we intend to make this clearer – these variables are definitely correlated, and we do not think they can be disentangled further without additional measurements (if at all).

Line 186: is west, NW, and SW 79%? That's missing from the sentence. Reporting 21% for the last source and not saying anything about the other 3 directions is reads strangely and compared to the prior sentence.

Thank you for pointing out this. We will clarify. Solution: This should read: In winter, 79% of the air mass back trajectories (n = 332) were from the west, NW, and SW, while 21% of air masses had backward trajectories south of the Falklands near the Antarctic Peninsula (Fig. S4).

Line 206-207: I think you need to be really careful presenting this here and then in the discussion below. With this data, maybe the other factors have a stronger control than precipitation d18O, but at least at some level, precip d18O must be important. So, when applying this down core (through Holocene), if there are changes in d18O, they must change the cellulose d18O (and then it's probably modified by the other factors you report here). I think this is critical to point out for those who will use this in paleo applications. More on this below.

We agree. Solution: We will address this more clearly and consistently in the text, but also in a new section we propose on paleo applications (see comments above) to make

this clear.

Section 3.4: there's no mention here of the relationships between the isotopes and temp and humidity but these are in the discussion, figures and tables. This would be a good place to describe the relationships of to both environmental factors.

This is addressed in section 3.5.

Line 237: What negative correlation? Not in the results or the figures. VPD is not discussed prior to this.

Thank you for the suggested change. It should read "positive". Solution: We will make the following changes: "Ferrio and Foltas (2005) established a positive correlation between $\delta$13Cleaf and vapor pressure deficit, suggesting stomatal conductance is sensitive to atmospheric moisture conditions."

Line 242-244: Is this consistent with the "low" humidity of the Falklands of >70%?

Yes, we will change this sentence to explain. Solution: "As plant stomata close in response to low humidity and/or high evaporative conditions like high wind speeds in the Falklands, the internal partial pressure of $CO_2$ decreases and the $\delta$13Cleaf increases (Farquhar et al., 1982, p.198)."

Lines 283-288: Relating plant tissue d18O (or dD) to precipitation is always a challenge. Even if you had leaf water or soil (pedestal?) water, it would still be complicated, but maybe give some insight. Many studies try to relate d18O of the plant back to precipitation, but here, it's clear that other factors modify this. But, at the most basic level, d18O precip is setting source water and then maybe there is mixing with other sources (ground water, dew, etc), but that is then modified by temp/humidity, etc. I think some discussion here is needed to highlight that this is much more complicated than indicated for the reader. If one tries to do this down core, changes in d18Oprecip must at some level matter for the d18O of plant source water and ultimately the d18O cellulose.

We agree. Thank you for pointing this out. Solution: We intend to address this in the Discussion by adding some sentences here to signal that this is not straightforward e.g., "Interpretations of d18O of leaf cellulose from downcore peat records would need to consider that the relationship with d18O in source water is confounded by relative humidity. Still, the d18O in plant water pools and d18O in leaf cellulose are primarily influenced by d18O in precipitation."

Also, getting into event precip (as mentioned) could be interesting, but it might be more informative to pull into this discussion when the leaves/cellulose are being made. Can you say anything about this with the data in hand?

No, we can not because we collected samples monthly. In the manuscript we described in our methods section that the youngest leaves of a new plant were collected each month. We assume leaves/cellulose are being made at this time (1 month) because Poa flabellata continuously grows. We will add that we assume the leaf cellulose is being made during the past 1 month of growth to the manuscript (line 128-129).

Solution: rewrite as "For leaf material, the inner developing (youngest) leaves were collected and assumed to represent the past month of growth."

Overall, the discussion is lacking a clear description of how the d13C and d18O would be used to interpret paleoclimate. Is it a temperature signal, a humidity signal, a source of precipitation signal? Or is it all of the above? How will a down core record be interpreted? Is there any way to put some uncertainty into this? How are you going to disentangle the multiple correlations between the isotopes themselves and the relationships with temp and humidity?

Without experimentation we can not disentangle temperature and humidity using d13C and d18O. We suggest in the manuscript that stomatal conductance is likely driving the changes in d13c and d18O because of relative humidity (Line 270-274). The precipitation signal can not be separated using these data.

Solution: We propose a new section paleoclimate interpretations where we address these peat core questions, as indicated above.

Figure 1: It would be nice here or elsewhere to have the wind diagrams and the precip source isotopes provided. I don't know what the figure limitations are for this journal, so maybe that's not possible. But, it sure would be nice to have a bit more of the great data collected here summarized in the main article figures.

Both of these are in the Supplemental Information (S2 and S4), will be moved to the main text if that would be helpful.

Figure 2: It would be nice if the interpretive strategy figure here was where that data is reported. The peat core is interesting, but not really discussed. It would be nice if it was to put into an interpretive strategy that could be used for downcore paleo Reconstructions.

We agree and would include this in the new proposed section (as commented above).

Figure 3: VSMOW on 3a, but VSMOW and VPDB missing on 3b. For the LMWL reported here, can you report the r or R2, p-value, and n? Figure 4, VSMOW and VPDB needed

These are all easy fixes, thank you.

———————————————————————

---

## Author Comment (AC2) · 13 Apr 2020

Dear Samuel Bodé,

We are grateful for your helpful comments, which have improved this manuscript. We have responded to your comments below. We also provide an explanation of the changes we intend to make in the manuscript.

Best wishes, Dulcinea Groff

*I believe this is an interesting piece of work, as indeed more reliable paleoclimatic proxies for the southern Atlantic are needed, to increase our understanding of past cli-

mate patterns. The author also collected a nice dataset. I do agree that Poa flabellata peat has promising potential to be the base of a good proxy, is it has a high accumulation rate in the peat, and is mainly present as the unique plant species. I do however not agree that the real poof of the power of the recorded isotopic signal as paleoclimate proxy has really been given in this manuscript. I have a couple of major concerns on the data treatment and interpretation and a large number of minor remarks.

First, the observed correlation of 13C and 18O of the leaf cellulose with RH and T is used as an indication of the power of the proxy for paleo climatic studies. The leaf samples were young leafs growing during the sampled year. The leaves start to grow vertically in the summer and get broader during the winter. The summer samples are thus systematically younger samples than the winter samples. It can not be excluded that the observed difference in 18O and 13C is related to the change in leaf phenology rather than a climatic response. As the entire leaves are collected in winter, the recorded isotopic signal is a combination of the entire growing season.

We believe the reviewer may have overlooked where we described our collection strategy. We collected the youngest leaves of a tussac plant each month to capture the most recent growth. We did not state in the manuscript that the leaves grow vertically in summer and broader during winter. Because we collected young inner leaf material we do not think the observed pattern is related to change in leaf phenology or systematically younger-older samples.

The time resolution of peat core reconstructions would average several years of accumulated plant matter in a single sample. When measuring 13C and 18O isotopes of cellulose from peat, any seasonal variation would be time averaged. Alternatively, the measured isotopes could suggest that past environmental conditions similar to or were from the seasonal relationship we observed in this study.

*Further, it is important to note that this seasonal resolution will not be recorded in the peat record, as only mature leaves will contribute to the litter. A much better way of

assessing the potential of the proxy for paleoclimatic reconstructions would of course be to sample a peat core, and correlate 13C and 18O signals of the core to recorded climate data. Therefore, it is needed to better frame the study, and rather use it as a background study on the physiological response of the Tussac grass and incorporation of atmospheric isotopic signal in the cellulose and only put it forward as a very first step toward the development of a paleo climatic proxy.

We would not assume that only mature leaves contribute to the litter. We think leaves are continuously growing and dying throughout the year.

Because of time averaging (∼20-30 years per cm) and limits to historic weather data [dating back to 1874; Lister and Jones 2015], it is not possible to get enough samples to establish a linkage between weather and isotopic values – we would have only 4-5 peat samples. We agree that paleo data would support this work with a proof of concept, but we wanted to focus on establishing the validity of the proxy through modern calibration, and we feel that these actualistic studies are important.

*Further comments: It is not always clear what is tested when statistical tests are performed. An example of this, is when the RH and T of the different locations are compared. It seems to me that the yearly average T and RH are compared using the individual days as replicates (I.e. SD computed on variation between days). This seems wrong to me as the variation in T and RH between individual days has no link with the uncertainty on the average T and RH of that location. To compare the RH and T of the measuring period for these locations only the measurement uncertainty (which is typically very small). Further to be able to say something on the difference in yearly average RH and T between sites in general, several years of observations are needed.

We are not trying to establish differences in weather conditions between sites, but rather we used multiple sites to test whether there was significant geographic variation in how tussac records T and RH. Our purpose was to show that these analyses can be done on cores from across the study region by establishing a link between T/RH and

leaf isotopes. Solution: We will clarify that the space for time substitution used here demonstrates that the processes influencing isotope signals in leaf cellulose observed over a season scales to inter annual change. We demonstrate that the geographic and seasonal patterns give us good confidence in the scalability of the relationship between leaf cellulose signals and humidity and temperature.

*The same problem occur when 18O and 2H in precipitation between seasons is compared, the monthly variability is not related to the uncertainty of the mean. On top of this, it is not so meaningful to compare the numerical average when comparing seasons, i.e. the weighted average should be used. Again only the measurement uncertainty is relevant when comparing the seasons. Using the weighed averages the average of the locations could be used to compare the different seasons, however, to me it doesn't seems right to use these different location as replicates.

It is unclear to us what the reviewer is requesting, so we are not sure how to respond. There may be some confusion about replicates; our sample sizes are similar between groups. Our use of multiple locations was to determine whether there was any spatial variation in stable isotope values, which would mean any paleoclimate work would require local sampling. That we found no such differences among site responses should be seen as a validation of this approach to the fossil record.

Further, it's important to note that our goal here is not to reconstruct seasonal changes, as peat records will include time-averaging of ∼20-30 years/cm.

*as a final note on this seasons start and end the 21th of a month, while samples were taken per month, this should also be acknowledged.) When looking at the correlation of the 13C and 18O in leaf cellulose with RH and T, it is observed that they correlate with both. Beside the issue discussed above it is also important to note that RH and T are also strongly correlated (i.e. Drier (77 -85 %)/hotter (9-11C) summers and wetter (90-98%)/colder (1-5.5C) winters). For which, from the data it can not be concluded if the effect is a result of the RH or T or both. It would be interesting to give some insight

on what effect might be prevailing here.

Thanks for taking note of this. We intend to acknowledge this. Solution: We intend to include a comment about the discretion between season (start/end on 21 of a month) and when samples were collected. For the second part of the comment, we would like to point out that we addressed this point on lines 270-274.

Smaller remarks: L75: it sounds quite contradictory to expect low WUE in water-limited environments.

Thank you for noticing this, it is an easy fix. Solution: Would be better if L75 read as, "environments where conservative water use strategies are common functional traits that allow. . ."

L80: It is a rather strange thing to say that 18O of source water often correlate with temperature. Better to say (and this is also how it is described in the given reference) that the 18O of precipitation correlates with temperature. (sure in the case of tussocks, source water is directly related to precipitation, but this cannot be claimed in general).

Agreed, better to revise this statement. Thank you. Solution: "Temperature of the environment is often correlated with $\delta$18O of source water (Libby et al., 1976).

L128: It is not clear what 'frozen for eight days' exactly means, what happened after these 8 days? In fact the entire section is not very clear, what is the point of freezing them for 8 days if some where already stored at RT for 6 months? Please rewrite and clarify.

Agreed, we will clarify. Solution: "Samples collected between September 2015 to February 2016 were frozen in February 2016 and samples collected in March 2016 to August/September 2016 were frozen in August/September 2016. Samples were frozen for eight days at the Falkland Islands Department of Agriculture to comply with U.S. Department of Agriculture permitting to prevent the spread of pests."

L144 and L150: When secondary reference material are used, the accepted value

used for it should be given (this do sometimes change over time).

Yes, we will add this. Solution: For L144: "...using USGS-42 (8.6 ‰ accepted value) and IAEA-601 cellulose (31.9‰ accepted value)...". For L150: "...included USGS-40 glutamic acid (-28.3‰ accepted value), USGS-41 glutamic acid (24.4 ‰ accepted value), and internal UWSIF $\alpha$-cellulose(-24.9 ‰ accepted value)."

L203: Paragraph is too long, to many irrelevant details are given for which the major lines get lost. I believe this paragraph should be shortened by c.a. 50%

Thank you for the suggestion. Solution: We will edit this paragraph.

L204: Why is this correlation analysis done on averaged values? It should be done using the individual data points.

Most correlation statistics assume independence of observations. Using individual data points instead of averaged values would violate this assumption when the goal is to compare seasonal variation across sites each month.

L13: replace 'investigate' by 'measured', delete 'plants'

Helpful suggestion, thank you. Solution: Rewrite as "Here, we measured the isotopic composition of Poa flabellata and..."

L14: I believe the author mean: '. . ... explore relationships with seasonal temperature and air humidity variations across 4. . ..'

Yes, thank you. Solution: We will change as described.

L16: Delete 'significantly' (if not you would not report it)

Good suggestion. Thank you.

L23: 'did not differ significantly' (there is no test to claim that 2 things do not differ, you can only claim that you could not see a significant difference.

Okay, good suggestion. We will change this. Solution: "No observed no significant

difference in the d18O values of monthly composite precipitation between seasons or study locations.

L32: '. . .resulted in an intensification and polward. . ..'

Sure, we will make this change. Solution: Change "correspond" to 'resulted'.

L35: Sentence starting with 'The inconsistency of. . ..' Is not totally clear, reformulate.

Thanks. We will fix by splitting into two sentences. Solution: "Meteorological measurements from the Falkland Islands date back to 1874 and are not continuous (Lister and Jones, 2015). This means we lack critical information on the long-term patterns and whether these are novel conditions."

L47: Should it not be '. . ..generate substantial amounts of peat. . .' ?

We will change this.

L57: I beleve it sould be cited as. 'Smith and Prince (1985) established radiocarbon. . .'

Yes, thank you, we will make this change.

L62: '. . ..of any peatland, globally, P. . .. . ."

Thanks. We will make this change.

L67: '. . ..in this semi-arid habitat. . .'

The references used here do not describe this semi-arid habitat in the Falkland Islands, these studies are about other places in the world that are semi-arid but not the Falklands.

L69: a) I guess it is 'up to 39 cm' or 'c.a. 39 cm' b) '. . .year) while in winter an increase. . ..'

We will make the recommended change. Solution: Change to "($\sim$ 39 cm per year)

while in winter..."

L70: '. . ..tiller at the base of. . ..'

Thank you for making this correction. Solution: Change to "...production of new tillers at the base of a. . ."

L70: Sentence starting with 'The climate signal. . ..' Is not clear, please reformulate.

Thank you, this sentence will be rewritten for clarity. Solution: Stable isotopes $\delta$18O, $\delta$D and $\delta$13C in the cellulose of plant tissues (roots, shoots, and leaves) can reliably record the climate signal related to environmental growing conditions (Araguás-Araguás et al., 2000). Carbon isotopes. . ."

L73: 'physiological responses such as changes in stomatal conductance and. . ..'

Nice suggestion. Will change accordingly.

L77: '. . .information on. . .'

Okay, sounds okay either way.

L78: '. . .tissue was formed. . .'

Okay.

L79: '. . .humidity and plant physiology (. . .'

Great suggestion. Solution: Change to "...temperature, humidity, and plant physiology (..."

L80: '. . .often correlate. . .'

Good suggestion. Will change. Solution: "...source water often correlates with temperature. . ."

L81: '. . .cellulose can also. . .'

Good suggestion. Solution: We will make this change. "....cellulose can also be influenced. . ."

L82: '. . ..plant water pools.'

Thank you. Solution: We will make the suggested change "...and plant water pools."

L86: rather use '‰$'

Good suggestion. Solution: This will be changed to be the first use of ‰ and first defined (removed first use later in the text L115-116).

L97: If mean temperature is given, the time span of this mean should be given, same for

Good point. We will add this. Solution: Our source references worldclim.org and we will add that the time span ranges from 1922-1988.

L109: Why is the fact that they were first shipped to University of Maine mentioned? I don't think the reader is interested In the postal rout. . ..

Okay. Solution: We will remove this, however it seems relevant to know how the samples were treated.

L113: It is '. . .were purified. . .' or 'Water was extracted out of precipitation sample. . ..'

Good suggestion. Solution: We will change to 'purified'.

L119: '. . . .relative to VSMOW.' No need to mention they are reported in ‰ (this is visible in the results, and can lead to inconsistency).

Good suggestion. Solution: Will make this change (it was also removed in a previous comment).

L130: '. . .were used, fine roots. . .'

Okay. We will remove the word "and" as suggested.

L133: Just out of personal curiosity, did the grinding of leaf material using a ball mill work? my experience is that this do not work very well with fibrous material.

The grinding of leaf material with the Retch ball mill worked very well. We used two balls in each canister to homogenize the fibrous material.

L137: a) What is an undetectable amount? Should give a detection limit here, if not it is meaningless. b) it is nitrogen and carbon content, not %nitrogen an %carbon

Agreed. Solution: This is how it could be re-written: "Further indicators of purity include low amounts of nitrogen content (0.13 % to 0.16 % N), and analysis of carbon content (42.1 % to 42.8 % C) in cellulose of four samples."

L140: 'varied by < 0.1 ‰ and 0.3 ‰ respectively'

Good suggestion. Solution: We will remove the plus/minus signs.

L143: '. . .relative to VSMOW (. . ..'

Good suggestion. Thank you for helping to tighten this language. Solution: We could rewrite as mentioned.

L146 and 151: replace ranges by 'c.a. 0.25 mg' and 'c.a. 2 mg'

Sure. We intend to add the targeted weight instead of the range of our weights. Solution: We will change this to reflect the target weights "c.a. 0.25" and "c.a. 2 mg".

L148: Delete '; units are expressed. . .. . .. . ..mil)'

Thanks for the suggestion. Solution: We will delete.

L150: Reformulate last sentence

L153: what is meant with 'analysing the average'? I think the author means that for every site 3 to 4 leafs were used as replicates in every month.

Our experimental unit was the average of 3-4 plant leaf samples (from new plants each time) at each site each month. We did not expect differences across sites. Solution: We will edit as suggested "...we averaged three to four plant leaf samples"

L184: what is the 'n = 344', summer only counts 90 days. . ..

Thank you for pointing this out. We agree this could use clarification. Solution: Rewrite like this: "...using n = 344 individual trajectories...(n = 332 individual trajectories) were from the..."

L189: Simply say that September 2015 sample was missing for surf bay (we all know that a year has 12 months). Was that sample missing or was it not sampled, meaning that October 2015 is in fact September + October?

Thank you, this definitely needs clarification. Solution: Rewrite: "...which was not sampled in September 2015, and October 2015 represents September and October 2015."

L192: '. . . D values (. . .' , 'monthly composite' is redundant with the first part of the sentence.

Great catch. Thank you, this will improve writing clarity. Solution: "D isotopes (y = . . .)."

L204: '. . ..leaf 18O and 13C values. . .'

Thank you. We will make this change to the symbols. Solution: "...monthly average leaf $\delta$13C and $\delta$18O values for..."

L206: add p value after segregation between winter and summer values.

Thanks. We will move our reported p-value to the end of the sentence. Solution: "...positive correlation (Pearson's r = 0.877, n = 24; Fig. 3B) and segregation between winter and summer values (p < 0.001)."

L206: How could measurement have a significant correlation? I believe the author

meant to write that the d18O value of precipitation did not correlate significantly. Further it would be more logical to write that d18O of leaf and root did not correlate with d18O of precipitation rather than the other way around (statistically it is the same thing, though it is not logical).

Sure. We will change to make it more logical, if that helps. Solution: "The $\delta$18Oleaf or $\delta$18Oroot did not correlate with $\delta$18O in precipitation across all..."

L212: '28.9 $\pm$ 1.3 ‰
'ıdem at L214 etc. . ..(change everywhere)

L213: I do not find it meaning full to add the range, as the distribution was normal, giving average $\pm$ SD is enough (and really no reason to add also the range).

We think both the standard deviation and the range are useful. Reporting the range is valuable for understanding patterns in the data, e.g., boundaries.

L222: interaction is not an effect, what the author wanted to say is that 'no significant interaction could be observed'.

Thanks for making this language clearer. We will make this change.

L237: Not clear if this is an own observation (nothing is mentioned about VPD in the result section) or something from literature.

Thanks for pointing out the need to clarify. Solution: We will clarify by writing "The observed negative correlation between d13C and relative humidity...conditions." instead of "The negative correlation...(Ferrio and Voltas 2004)."

L241: '. . .ratio of CO2 partial pressure in the leaf and that of the ambient air. . .'

Great, thanks. We like this suggestion and will rewrite. Solution: "...ratio of CO2 partial pressure in the leaf and that of the ambient air..."

L241: what can be explained, the difference in parial pressure ratio or the effect of it

on the 13C?

Thanks for noting this. We will clarify. Solution: We will rewrite like this "Variation in d13C is driven by changes...2000)."

L257: what are 'cellulose isotopes'?

Thanks. Solution: We will remove the word "isotopes" and the sentence reads well.

L317: Quite strange to say 'at least 12,500 14C years, while on line 64 it says that peatlands initiated between 12,500 and 5,500 14C years. . ..

Thank you for pointing this out. This sentence could use clarification. Solution: We could be consistent and say "some peatlands initiated by 12,500 14C years. . ." instead of giving a range.

Figure 3b: Why is not the individual data presented, rather than averages?

We designed the experiment so that each replicate is one month at each site. In other words, each experimental unit is an average of 3-4 isotope measurements from one site (four total sites).

Table S1: if the it is given that longitude is south, the negative sign should not be used (sensu stricto a negative south latitiude is a northen latitude). So remove the '(S)' or the '-' idem for long (W).

Great suggestion. Thank you for your attention to detail. Solution: We will remove the "-" from provided latitude/longitude.

Figure S3: Link did not work, until I found that the '-' was not for a split for a line break (like my browser interpreted it when clicking on it), but a real hyphen. Probably better to use 'https://climatereanalyzer.org/'

Okay, thank you. Solution: We will update with the new website name.

Figure S5: Add your data to this graph.

Good suggestion. Solution: We will add our data to this graph. However, this would mean the data would be presented twice: once in the main manuscript and once in the supplemental information.
* * *

---

## Author Response (AR1)

Dear Anonymous Reviewer,

We are grateful for the comments you provided, which have greatly improved this manuscript. We addressed each of your comments (blue italic text) and provide explanations of the changes we made.

Kindly,
Dulcinea Groff

*There is a lot of analysis into explaining the variation in the isotopes, how that's controlled by plant physiology, but not much discussion and explanation of how these isotope signals will be used to reconstruct paleoclimate especially in context of applying this to a peat core (through time). Providing a roadmap for how changes in d13C and d18O will be interpreted would be useful and weather this is qualitative or can it be pushed further to be quantitative?*

Peat-based reconstructions may be limited to identifying periods of warm/dry or cold/wet conditions that are more extreme than our observed seasonal variations (or more similar to them). For now, this proxy remains qualitative, but more work could be done to evaluate this proxy to assess its suitability for quantitative reconstructions (perhaps with leaf wax alkenones, though our preliminary data on hydrogen isotopes in precipitation suggests this may not be feasible). Resolving temperature and moisture signals independently would likely require growth chamber experiments.

**Solution:** We agree and added an expanded discussion on paleoclimate reconstructions (Lines 303-337) with a conceptual figure (Fig. 8) outlining what this proxy can reconstruct, including time-averaging and methods (i.e. analysis of bulk, individual leaves or roots).

*There is the suggestion that this is going to really help us understand climate dynamics, but then there is not discussion of how. Is this going to provide temperature or relative humidity or both (there is not clear indication of which and both are correlated with the isotopes) and how to you disentangle any changes in source water d18O through time?*

The oxygen and carbon isotopic signatures are positively correlated in our study, which represents a modern snapshot of conditions. Based on this modern calibration work, we recommend measuring C and O values together, and that these values would identify the timing of transitions between warmer and drier to colder and wetter conditions that correlate with major hemispheric drivers in climate.

We didn't find a relationship between d18O in precipitation and the d18O of leaf cellulose in our 1-year study. It is possible that because the d18O in leaf water is controlled by source water and humidity, any changes in humidity confound a direct relationship between source water d18O values and leaf water/cellulose d18O. It may also be possible that the variation in source area did not affect the d18O enough for us to detect a significant impact on the d18O of the leaf cellulose. Based on this work, the source area could influence leaf cellulose d18O value in peat records if source area changes were greater than what we observed.

**Solution:** Thank you for pointing out that this is unclear in the manuscript. We added a section to the Discussion (Line 303-337) that more clearly outlines how this proxy would work in practice, including the conceptual figure (Fig. 8), and clarification about the disentangling source

water d18O through time.

*Some discussion of how to do this for paleoclimate also needs to focus on how this study shows nicely that the leaves are recording a seasonal signal. So, when you go down core, how are you going to deal with this? Are you going to focus on a large sampling of leaves from each horizon (age?) with the expectation that you are sampling both seasons or is it going to be a single multiple leaf measurement to approximate an annual signal? Some thought into this is needed as the data analysis and presentation may need to be added to or adapted for paleo work. I'd like to see a clearer connection between this nice modern calibration data and how to use it for the past.*

We recommend incorporating a large sampling of leaves from a horizon (1 cm), and expect that to be a sampling of both seasons, incorporating several years. These peat records are highly productive, but even so, time averaging within a given 1-cm level should be greater than than the age of a leaf. In the peat records we've analyzed, sedimentation rates can be ~20-30 years/cm, with most records extending at least 12,500 years old. As with any prehistoric reconstruction, it is important to consider temporal grain and resolution, which is going to constrain the inferences you can make.

**Solution**: Thank you for pointing out the need for this additional discussion. We included new text in the Discussion section (Lines 303-337) and conceptual figure (Fig. 8) with a recommended workflow and considerations for applying to peat records, including our recommendations for methods (e.g., including that multiple whole leaf fragments should be used from each level).

*Line 28: "trends in southern hemisphere climate dynamics" – is that consistent with what you can actually do with this proxy? Or is it something more specific?*

Based on what we have established with this study, this proxy can indicate trends in conditions similar to what we observed seasonally: warm/dry, cool/wet, which is more specific than our generalized statement. Multiple paleo-records could point to changes in Southern Hemisphere climate dynamics, but can not necessarily resolve the drivers of those dynamics.

*Lines 43-46: Awkward sentence with semicolon connecting two separate statements.*

Thanks for pointing this out. **Solution:** We removed the semicolon and edited for clarity (Lines 44-45).

*Line 56: Is it really called a "bog"? That's not confusing... It's hard to reconcile this description with the one line 70 and "pedestal" which is in the caption for Figure 2. Maybe some annotation on the figure or more description would be useful. I'd like to have a clear idea how this is going to develop over time in a peatland and how this plants growth habit is going to translate into a vertical succession (or some crazy patchwork of different ages in a peat core).*

Yes, colloquially each pedestal is called a "bog," and we removed this from the manuscript to avoid confusion -- especially because the tussac peat that forms the soils in these stands is not a bog, either. We called it a "pedestal" throughout. ("Tussac/tussock" is already confusing enough.)

The taphonomy of these pedestals is poorly understood. There could be a patchwork of ages within a peatland, but as of yet we have not found any evidence of age reversals or other chronology problems in our cores.

*Line 57: Something wrong with new sentence that starts here and sentence seems incomplete too.*

Thank you for pointing this out. **Solution:** We edited the sentence as "Smith and Prince (1985) established radiocarbon…" on Line 57.

*Line 56-57: Either they use precip or the precip wets all that organic matter and then there is evaporative enrichment b/c it is exposed to wind/sun.*

Fair point. **Solution:** We edited the text on Line 56 to indicate this.

*Line 70: Maybe start a new paragraph here or have a better transition?*

Thank you for this suggestion. We started a new paragraph and improved the opening sentence on Lines 70-73.

*Line 71 and below: check the order in which isotopes are first described. Here delta symbols are used first but aren't defined, next sentence doesn't use delta symbols (carbon isotopes), and then defined on line 90-91. I think this comes up a few other places and would be worth cleaning up.*

Thanks for noting this. We fixed this for consistency throughout.

*Lines 92-93: Improving "westerly wind dynamics" is different than what's mentioned elsewhere. What is it that this new proxy can solve and make it consistent throughout.*

Great point. We made the text clearer regarding exactly what climate conditions and questions this  proxy can inform, while still placing it within the broader discussion of Southern Hemisphere climate dynamics. We used "Southern Hemisphere climate dynamics" on Line 93 and remained consistent throughout.

*Line 100: Could the km hr-1 also be reported here and later for reference? Not to many readers will think about wind speed in m/s.*

The International System of Units recommends m/s, so we will retain use of this standard for consistency with them, and with other studies (https://physics.nist.gov/cuu/Units/units.html). **Solution**: We added km/h in parenthesis after the recommended m/s (Lines 179, 181, and 183).

*Line 170-172: How are temperature and humidity related? Based on the figure, they look highly correlated. If they are, then how do you disentangle their effects from the cellulose d13C and d18O as they are both strongly related? I didn't see any multiple regression analysis reported below either.*

As discussed in the new section in the Discussion, we made this clearer (Lines 303-337). These variables are definitely correlated, and we do not think they can be disentangled further without additional measurements.

*Line 186: is west, NW, and SW 79%? That's missing from the sentence. Reporting 21% for the last source and not saying anything about the other 3 directions is reads strangely and compared to the prior sentence.*

Thank you for pointing out this missing reported value. **Solution:** This is now re-written on Lines 186-188: "In winter, 79% of the air mass back trajectories (n = 332 individual trajectories) were from the west, NW, and SW, while 21% of air masses had backward trajectories south of…".

*Line 206-207: I think you need to be really careful presenting this here and then in the discussion below. With this data, maybe the other factors have a stronger control than precipitation d18O, but at least at some level, precip d18O must be important. So, when applying this down core (through Holocene), if there are changes in d18O, they must change the cellulose d18O (and then it's probably modified by the other factors you report here). I think this is critical to point out for those who will use this in paleo applications. More on this below.*

We agree. **Solution:** We addressed this more clearly in the new section of the Discussion (Lines 303-337) and conceptual figure (Fig. 8) on paleo applications to make this clear. We rephrased the sentence using clearer language on Lines 208-209: "The $\delta 18 O_{leaf}$ or $\delta 18 O_{root}$ did not correlate with $\delta 18 O$ in precipitation across all sites (Table 1)."

*Section 3.4: there's no mention here of the relationships between the isotopes and temp and humidity but these are in the discussion, figures and tables. This would be a good place to describe the relationships of to both environmental factors.*

This is addressed in section 3.5.

*Line 237: What negative correlation? Not in the results or the figures. VPD is not discussed prior to this.*

Thank you for identifying this mistake. It now reads "positive". **Solution:** We made the following changes on Line 239: "Ferrio and Voltas (2005) established a positive correlation between $\delta 13 C_{leaf}$ and vapor pressure deficit, suggesting stomatal conductance is sensitive to atmospheric moisture conditions."

*Line 242-244: Is this consistent with the "low" humidity of the Falklands of >70%?*

Yes, we changed this sentence to explain on Lines 245-247. **Solution:** "As plant stomata close in response to low humidity and/or high evaporative conditions like high wind speeds in the Falklands, the internal partial pressure of $CO_2$ decreases and the $\delta 13 C_{leaf}$ increases (Farquhar et al., 1982, p.198)."

*Lines 283-288: Relating plant tissue d18O (or dD) to precipitation is always a challenge. Even if you had leaf water or soil (pedestal?) water, it would still be complicated, but maybe give some insight. Many studies try to relate d18O of the plant back to precipitation, but here, it's clear that other factors modify this. But, at the most basic level, d18O precip is setting source water and then maybe there is mixing with other sources (ground water, dew, etc), but that is then modified by temp/humidity, etc. I think some discussion here is needed to highlight that this is much more complicated than indicated for the reader. If one tries to do this down core, changes*

*in d18Oprecip must at some level matter for the d18O of plant source water and ultimately the d18O cellulose.*

We agree. Thank you for pointing this out. **Solution:** We addressed this in the Discussion by adding sentences here to signal that this is not straightforward on Lines 303-337.

*Also, getting into event precip (as mentioned) could be interesting, but it might be more informative to pull into this discussion when the leaves/cellulose are being made. Can you say anything about this with the data in hand?*

No, we can not because we collected samples monthly. In the manuscript we described in our methods section that the youngest leaves of a new plant were collected each month. We assume leaves/cellulose are being made at this time (1 month) because Poa flabellata continuously grows.

**Solution:** We added that we assume the leaf cellulose is being made during the past 1 month of growth to the manuscript on Lines 128-129.

*Overall, the discussion is lacking a clear description of how the d13C and d18O would be used to interpret paleoclimate. Is it a temperature signal, a humidity signal, a source of precipitation signal? Or is it all of the above? How will a down core record be interpreted? Is there any way to put some uncertainty into this? How are you going to disentangle the multiple correlations between the isotopes themselves and the relationships with temp and humidity?*

Without experimentation we can not disentangle temperature and humidity using d13C and d18O. We suggest in the manuscript that stomatal conductance is likely driving the changes in d13c and d18O because of relative humidity (Lines 272-273). The precipitation signal can not be separated using these data.

**Solution**: We address these peat core questions in the new section about paleoclimate interpretations (Lines 303-337).

*Figure 1: It would be nice here or elsewhere to have the wind diagrams and the precip source isotopes provided. I don't know what the figure limitations are for this journal, so maybe that's not possible. But, it sure would be nice to have a bit more of the great data collected here summarized in the main article figures.*

Great idea, thank you. We moved both supplemental figures to the main text, as suggested.

*Figure 2: It would be nice if the interpretive strategy figure here was where that data is reported. The peat core is interesting, but not really discussed. It would be nice if it was to put into an interpretive strategy that could be used for downcore paleo Reconstructions.*

We agree and included this in the new section in the Discussion on Lines 303-337and conceptual figure (Fig. 8).

*Figure 3: VSMOW on 3a, but VSMOW and VPDB missing on 3b. For the LMWL reported here, can you report the r or R2, p-value, and n? Figure 4, VSMOW and VPDB needed*

These were all fixed, thank you.

Dear Samuel Bodé,

Thank you for your helpful comments and edits, which have improved this manuscript. We responded to each of your points (blue italic text by providing an explanation of the changes we made in the manuscript.

Kindly,
Dulcinea Groff

*I believe this is an interesting piece of work, as indeed more reliable paleoclimatic proxies for the southern Atlantic are needed, to increase our understanding of past climate patterns. The author also collected a nice dataset. I do agree that Poa flabellate peat has promising potential to be the base of a good proxy, is it has a high accumulation rate in the peat, and is mainly present as the unique plant species. I do however not agree that the real poof of the power of the recorded isotopic signal as paleoclimate proxy has really been given in this manuscript. I have a couple of major concerns on the data treatment and interpretation and a large number of minor remarks.*

*First, the observed correlation of 13C and 18O of the leaf cellulose with RH and T is used as an indication of the power of the proxy for paleo climatic studies. The leaf samples were young leafs growing during the sampled year. The leaves start to grow vertically in the summer and get broader during the winter. The summer samples are thus systematically younger samples than the winter samples. It can not be excluded that the observed difference in 18O and 13C is related to the change in leaf phenology rather than a climatic response. As the entire leaves are collected in winter, the recorded isotopic signal is a combination of the entire growing season.*

We described our collection strategy on Lines 125-129. We collected the youngest leaves of a tussac plant each month to capture the most recent growth (it continuously grows). Because we collected young inner leaf material we do not think the observed pattern is related to change in leaf phenology or systematically younger-older samples.

The time resolution of peat core reconstructions would average several years of accumulated plant matter in a single sample. When measuring d13C and d18O isotopes of cellulose from peat, any seasonal variation would be time-averaged. Alternatively, the measured isotopes could suggest that past environmental conditions were similar to or different from the seasonal relationship we observed in this study. We discuss this point in our new text in the Discussion section on Lines 303-337and in a conceptual figure (Fig. 8).

*Further, it is important to note that this seasonal resolution will not be recorded in the peat record, as only mature leaves will contribute to the litter. A much better way of assessing the potential of the proxy for paleoclimatic reconstructions would of course be to sample a peat core, and correlate 13C and 18O signals of the core to recorded climate data. Therefore, it is needed to better frame the study, and rather use it as a background study on the physiological response of the Tussac grass and incorporation of atmospheric isotopic signal in the cellulose and only put it forward as a very first step toward the development of a paleo climatic proxy.*

We would not assume that only mature leaves contribute to the litter. We think leaves are continuously growing and dying throughout the year.

Because of time averaging (~20-30 years / cm) and limits to historic weather data [dating back to 1874; Lister and Jones 2015], it is not possible to get enough samples to establish a linkage between weather and isotopic values -- we would have only 4-5 peat samples. We agree that paleo data would support this work with a proof of concept, but we wanted to focus on establishing the validity of the proxy through modern calibration, and we think that these actualistic studies are important.

*Further comments: It is not always clear what is tested when statistical tests are performed. An example of this, is when the RH and T of the different locations are compared. It seems to me that the yearly average T and RH are compared using the individual days as replicates (I.e. SD computed on variation between days). This seems wrong to me as the variation in T and RH between individual days has no link with the uncertainty on the average T and RH of that location. To compare the RH and T of the measuring period for these locations only the measurement uncertainty (which is typically very small). Further to be able to say something on the difference in yearly average RH and T between sites in general, several years of observations are needed.*

We are not trying to establish differences in weather conditions between sites, but rather we used multiple sites to test whether there was significant geographic variation in how tussac records T and RH. Our purpose was to show that these analyses can be done on cores from across the study region by establishing a link between T/RH and leaf isotopes.

**Solution:** The new text in Discussion (Lines 303-337) and conceptual figure (Fig. 8) clarify that the space for time substitution used here demonstrates that the processes influencing isotope signals in leaf cellulose observed over a season scales to inter annual change. We demonstrate that the geographic and seasonal patterns give us confidence in the scalability of the relationship between leaf cellulose signals and humidity and temperature.

*The same problem occur when 18O and 2H in precipitation between seasons is compared, the monthly variability is not related to the uncertainty of the mean. On top of this, it is not so meaningful to compare the numerical average when comparing seasons, i.e. the weighted average should be used. Again only the measurement uncertainty is relevant when comparing the seasons. Using the weighed averages the average of the locations could be used to compare the different seasons, however, to me it doesn't seems right to use these different location as replicates.*

It is unclear to us what the reviewer is requesting, so we are not sure how to respond. There may be some confusion about replicates. Our sample sizes are similar between groups. Our use of multiple locations was to determine whether there was any spatial variation in stable isotope values, which would mean any paleoclimate work would require local sampling. That we found no such differences among site responses should be seen as a validation of this approach to the fossil record. We noted in our conceptual figure and new text in the Discussion section (Lines 303-337) that our goal here is not to reconstruct seasonal changes, as peat records will include time-averaging of ~20-30 years/cm.

*as a final note on this seasons start and end the 21th of a month, while samples were taken per month, this should also be acknowledged.) When looking at the correlation of the 13C and 18O in leaf cellulose with RH and T, it is observed that they correlate with both. Beside the issue discussed above it is also important to note that RH and T are also strongly correlated (i.e. Drier (77 -85 %)/hotter (9-11C) summers and wetter (90-98%)/colder (1-5.5C) winters). For which,*

*from the data it can not be concluded if the effect is a result of the RH or T or both. It would be interesting to give some insight on what effect might be prevailing here.*

Thanks for taking note of this. **Solution:** We acknowledged this with a new sentence on Lines 155-156: "Because samples were collected at the start of each month, we define summer as the months of DJF and winter as the months JJA."

For the second part of the comment, we would like to point out that we addressed this point on Lines 269-275.

*Smaller remarks:*
*L75: it sounds quite contradictory to expect low WUE in water-limited environments.*

Thank you for noticing. **Solution**: We edited the sentence to read as, "environments where conservative water use strategies are common functional traits that allow…" on Lines 74-75.

*L80: It is a rather strange thing to say that 18O of source water often correlate with temperature. Better to say (and this is also how it is described in the given reference) that the 18O of precipitation correlates with temperature. (sure in the case of tussocks, source water is directly related to precipitation, but this cannot be claimed in general).*

Agreed, the statement is better revised. Thank you. **Solution:** "The δ18O of source water often correlates with temperature of the environment (Libby et al., 1976)." Lines 79-80."

*L128: It is not clear what 'frozen for eight days' exactly means, what happened after these 8 days? In fact the entire section is not very clear, what is the point of freezing them for 8 days if some where already stored at RT for 6 months? Please rewrite and clarify.*

Agreed, we edited for clarification. **Solution:** "Samples collected between September 2015 to February 2016 were frozen in February 2016 and samples collected in March 2016 to August/September 2016 were frozen in August/September 2016. Samples were frozen for eight days at the Falkland Islands Department of Agriculture to comply with U.S. Department of Agriculture permitting to prevent the spread of pests." (Lines 125-128).

*L144 and L150: When secondary reference material are used, the accepted value used for it should be given (this do sometimes change over time).*

Thank you. **Solution**: We added this information to Lines 144-150:

"...using USGS-42 (8.6 ‰ accepted value) and IAEA-601 cellulose (31.9‰ accepted value)..."

"...included USGS-40 glutamic acid (-28.3‰ accepted value), USGS-41 glutamic acid (24.4 ‰ accepted value), and internal UWSIF α-cellulose (-24.9 ‰ accepted value)."

*L203: Paragraph is too long, to many irrelevant details are given for which the major lines get lost. I believe this paragraph should be shortened by c.a. 50%*

It is not clear what is being suggested.

*L204: Why is this correlation analysis done on averaged values? It should be done using the individual data points.*

Most correlation statistics assume independence of observations. Using individual data points instead of averaged values would violate this assumption when the goal is to compare seasonal variation across sites each month.

*L13: replace 'investigate' by 'measured', delete 'plants'*

Helpful suggestion, thank you. **Solution:** Rewritten as "Here, we measured the isotopic composition of Poa flabellata and…" on Line 13.

*L14: I believe the author mean: '. . ... explore relationships with seasonal temperature and air humidity variations across 4. . ..'*

Yes, thank you. **Solution:** We changed as described on Line 14.

*L16: Delete 'significantly' (if not you would not report it)*

Good suggestion. We made the change on Line 16..

*L23: 'did not differ significantly' (there is no test to claim that 2 things do not differ, you can only claim that you could not see a significant difference.*

Okay, good suggestion. **Solution:** We changed the sentence to "$\delta^{18}O$ values of monthly composite precipitation were similar between seasons or among study locations, yet characteristic of the latitudinal origin of storm tracks and seasonal winds." on Lines 23-24.

*L32: '. . .resulted in an intensification and polward. . ..'*

Okay, we made this change. **Solution**: Changed "correspond" to 'resulted' on Line 32.

*L35: Sentence starting with 'The inconsistency of. . ..' Is not totally clear, reformulate.*

Thanks. We clarified and reformulated by splitting into two sentences. **Solution:** "Meteorological measurements from the Falkland Islands date back to 1874 and are not continuous (Lister and Jones, 2015). This means we lack critical information on the long-term patterns and whether these are novel conditions." on Lines 35-37.

*L47: Should it not be '. . ..generate substantial amounts of peat. . .' ?*

Thanks. We changed this on Line 46 exactly as suggested.

*L57: I beleve it sould be cited as. 'Smith and Prince (1985) established radiocarbon. . .'*

Yes, thank you, we made this change on Line 57.

*L62: '. . ..of any peatland, globally, P. . .. . .."*

Thanks. We made this change on Line 62.

*L67: '. . ..in this semi-arid habitat. . .'*

The references used here do not describe this semi-arid habitat in the Falkland Islands, these studies are about other places in the world that are semi-arid but not the Falkland Islands.

*L69: a) I guess it is 'up to 39 cm' or 'c.a. 39 cm' b) '. . .year) while in winter an increase. . .'*

We made the recommended change. **Solution**: Changed to "(~ 39 cm per year) while in winter..." on Line 68.

*L70: '. . ..tiller at the base of. . ..'*

Thank you for making this suggestion. **Solution**: Changed to "...production of new tillers at the base of a…" on Line 69.

*L70: Sentence starting with 'The climate signal. . ..' Is not clear, please reformulate.*

Thank you, this sentence was rewritten for clarity. We started a new paragraph here and revised the sentence. **Solution: "**Stable isotopes δ18O, δD and δ13C in the cellulose of plant tissues (roots, shoots, and leaves) can reliably record the climate signal related to environmental growing conditions (Araguás-Araguás et al., 2000)." on Line 70-72.

*L73: 'physiological responses such as changes in stomatal conductance and. . ..'*

Nice suggestion. Will changed accordingly on Lines 73-74.

*L77: '. . .information on. . .'*

Okay, sounds okay either way on Line 77.

*L78: '. . .tissue was formed. . .'*

Okay, change was made on Line 77.

*L79: '. . .humidity and plant physiology (. . .'*

Great suggestion. **Solution**: Changed to "...temperature, humidity, and plant physiology (..." on Line 79.

*L80: '. . .often correlate. . .'*

Good suggestion. . **Solution**: We made the change "...source water often correlates with temperature…" on Line 80.

*L81: '. . .cellulose can also. . .'*

Good suggestion. **Solution:** We made this change. "....cellulose can also be influenced…" on Line 81.

*L82: '. . ..plant water pools.'*

Thank you. **Solution**: We made the suggested change "...and plant water pools." on Line 81.

*L86: rather use '‰´*

Good suggestion. We added "(‰)" at first use  on Line 86.

*L97: If mean temperature is given, the time span of this mean should be given, same for*

Good point. **Solution**: Our source (Turner and Pendelbury 2000) references worldclim.org and we added that the time span ranges from 1922-1988 on Line 97.

*L109: Why is the fact that they were first shipped to University of Maine mentioned? I don't think the reader is interested In the postal rout. . ..*

Okay. **Solution**: We removed this, however it seems relevant to know how the samples were treated.

*L113: It is '. . .were purified. . .' or 'Water was extracted out of precipitation sample. . ..'*

Good suggestion. **Solution**: We changed to 'purified' on Line 113.

*L119: '. . .relative to VSMOW.' No need to mention they are reported in ‰ (this is visible in the results, and can lead to inconsistency).*

Good suggestion. **Solution:** We made this change on Line 118.

*L130: '. . .were used, fine roots. . .'*

Okay. We removed the word "and" as suggested on Lines 131-132.

*L133: Just out of personal curiosity, did the grinding of leaf material using a ball mill work? my experience is that this do not work very well with fibrous material.*

The grinding of leaf material with the Retch ball mill worked very well. We used two balls (~ 1 cm diameter ball) in each canister to homogenize the fibrous material.

*L137: a) What is an undetectable amount? Should give a detection limit here, if not it is meaningless. b) it is nitrogen and carbon content, not %nitrogen an %carbon*

Agreed. **Solution**: We edited the sentence like this: "Further indicators of purity include low amounts of nitrogen content (0.13 % to 0.16 % N), and analysis of carbon content (42.1 % to 42.8 % C) in cellulose." on Lines 136-138.

*L140: 'varied by < 0.1 ‰ and 0.3 ‰ respectively'*

Good suggestion. **Solution**: We removed the plus/minus signs on Line 140.

*L143: '. . .relative to VSMOW (. . ..'*

Good suggestion. Thank you for helping to tighten this language. **Solution:** The sentence was rewritten as suggested on Line 143.

*L146 and 151: replace ranges by 'c.a. 0.25 mg' and 'c.a. 2 mg'*

Sure. **Solution**: The sentence was edited to reflect the target weigh instead of the range of our weights "c.a. 0.25" and "c.a. 2 mg" on Line 146 and Line 151.

*L148: Delete '; units are expressed. . .. .. . ..mil)'*

Thanks for the suggestion. **Solution:** We deleted this.

*L150: Reformulate last sentence*

*L153: what is meant with 'analysing the average'? I think the author means that for every site 3 to 4 leafs were used as replicates in every month.*

Our experimental unit was the average of 3-4 plant leaf samples (from new plants each time) at each site each month. We did not expect differences across sites. **Solution**: We edited as suggested "...we averaged three to four plant leaf samples" on Line 154.

*L184: what is the 'n = 344', summer only counts 90 days. . ..*

Thank you for pointing this out. We agree this needed clarification. **Solution**: The sentence was rewritten like this: "...using n = 344 individual trajectories…(n = 332 individual trajectories) were from the…" on Lines 186-188

*L189: Simply say that September 2015 sample was missing for surf bay (we all know that a year has 12 months). Was that sample missing or was it not sampled, meaning that October 2015 is in fact September + October?*

Thank you, this definitely needed clarification. **Solution**: We edited the sentence: "...which was not sampled in September 2015, and October 2015 represents September and October 2015." on Lines 195-196.

*L192: '. . . D values (. . .' , 'monthly composite' is redundant with the first part of the sentence.*

Great catch. Thank you, this improved writing clarity. **Solution**: "D isotopes (y = …)." on Lines 191-192.

*L204: '. . ..leaf 18O and 13C values. . .'*

Thank you. **Solution**: We removed "stable isotopes" so now it reads as"...monthly average leaf δ13C and δ18O values for…" on Lines 208-209.

*L206: add p value after segregation between winter and summer values.*

Thanks. **Solution**: We added our reported p-value from below (Line 216), p < 0.001, to the end of the sentence (Line 208).
 "...positive correlation (Pearson's r = 0.877, p < 0.001, n = 24; Fig. 3B) and segregation between winter and summer values (p < 0.001)." on Line 208.

*L206: How could measurement have a significant correlation? I believe the author meant to write that the d18O value of precipitation did not correlate significantly. Further it would be more logical to write that d18O of leaf and root did not correlate with d18O of precipitation rather than the other way around (statistically it is the same thing, though it is not logical).*

Okay. **Solution**: We changed this to make it more logical:  "The δ18Oleaf or δ18Oroot did not correlate with δ18O in precipitation across all…" on Line 208-209.

*L212: '28.9 ± 1.3 ‰ídem at*

*L214 etc. . ..(change everywhere)*

We made these changes. Thanks.

*L213: I do not find it meaning full to add the range, as the distribution was normal, giving average ± SD is enough (and really no reason to add also the range).*

We think both the standard deviation and the range are useful. Reporting the range is valuable for understanding patterns in the data, e.g., boundaries.

*L222: interaction is not an effect, what the author wanted to say is that 'no significant interaction could be observed'.*

Thanks for making this language clearer. **Solution**: We made this change on Lines 219-220.

*L237: Not clear if this is an own observation (nothing is mentioned about VPD in the result section) or something from literature.*

Thanks for pointing out the need to clarify. **Solution:** We clarified by rewriting to indicate it is information from literature on Line 239.

*L241: '. . .ratio of CO2 partial pressure in the leaf and that of the ambient air. . ..'*

Great, thanks. **Solution**: We like this suggestion and edited the sentence like this: "...ratio of CO2 partial pressure in the leaf and that of the ambient air…" Line 243.

*L241: what can be explained, the difference in parial pressure ratio or the effect of it on the 13C?*

Thanks for noting this. We will clarify. **Solution**: We will rewrite like this "Variation in d13C is driven by changes...2000)." on Lines 242-243.

*L257: what are 'cellulose isotopes'?*

Thanks. **Solution**: We removed "isotopes" on Line 257 and the sentence reads well.

*L317: Quite strange to say 'at least 12,500 14C years, while on line 64 it says that peatlands initiated between 12,500 and 5,500 14C years. . ..*

Thank you for pointing this out. This sentence could use clarification. **Solution**: We rewrote the first sentence on (Line 63) for consistency (instead of range) with the text "and date back to at least 12,500 14C years,…" found on Line 343.

"…peatlands initiated by 12,500 14C years…" on Line 63

*Figure 3b: Why is not the individual data presented, rather than averages?*

We designed the experiment so that each replicate is one month at each site. In other words, each experimental unit is an average of 3-4 isotope measurements from one site (four total sites).

*Table S1: if the it is given that longitude is south, the negative sign should not be used (sensu stricto a negative south latitiude is a northen latitude). So remove the '(S)' or the '-' idem for long (W).*

Great suggestion. Thank you for your attention to detail. **Solution**: We removed the "-" from provided latitudes.

*Figure S3: Link did not work, until I found that the '-' was not for a split for a line break (like my browser interpreted it when clicking on it), but a real hyphen. Probably better to use 'https://climatereanalyzer.org/'*

Okay, thank you. **Solution**: We updated with the new website name:
https://climatereanalyzer.org/

*Figure S5: Add your data to this graph.*

Good suggestion. **Solution**: Based on the suggestion by Reviewer 1, we moved these data to Fig 4. in the main text. The data seem to be best displayed in two panels rather than one plot.

[revised manuscript text omitted]

180    summer, >20% winds from the SW were between 5 and 10 m s$^{-1}$, and there was a higher frequency of 10 to 15 m s$^{-1}$ (36 to 54 km h$^{-1}$) wind speeds than in winter. Seasonal wind variation deviated from the long-term average (1979-2015). Reanalysis data (ERA Interim; Fig. S2) indicated that the wind speeds during summer (DJF 2015 to 2016) were stronger over the Falkland Islands (5 to 6 m s$^{-1}$) (18 to 21 km h$^{-1}$) and weaker during winter (JJA 2016).

**3.3 Seasonal HYSPLIT air mass trajectory analyses**

185    The daily back trajectory HYSPLIT analysis indicated that during the summer, 89% of the air masses originated ($n$ = 344 individual trajectories) west of the Falkland Islands. Approximately 11% of summer air masses originated south of the Falkland Islands near the Antarctic Peninsula. In winter, 79% of the air mass back trajectories ($n$ = 332 individual trajectories) were from the west, NW, and SW, while 21% of air masses had backward trajectories south of the Falkland Islands near the Antarctic Peninsula (Fig. 3).

190    **3.4 Monthly composite precipitation, δ$^{18}$O and δD**

Each study location had $n$ = 12 samples over the year, except for Surf Bay ($n$ = 11), which was not sampled in September 2015, and October 2015 represents September and October 2015. Monthly composite δ$^{18}$O and δD isotopes in precipitation throughout the year ranged from -12.3 ‰ to -4.8 ‰, and from -86 ‰ to -23 ‰, respectively. Monthly composite precipitation at each location was used to construct a local meteoric water line using δ$^{18}$O and δD isotopes

195    ($y$ = 7.571x + 5.527; $n$ = 47; Fig. 4a). The range for winter δ$^{18}$O and δD was from -8.6 ‰ to -6.6 ‰ and -61 ‰ to -40 ‰, respectively. Summer values of δ$^{18}$O and δD in precipitation ranged from -12.3 ‰ to -5.3 ‰ and -86 ‰ to -38 ‰, respectively, and fit within the range of historical isotopes in precipitation from the Falkland Islands (GNIP; Fig. 4b). Summer and winter δ$^{18}$O and δD isotopes in precipitation ($n$ = 24) passed tests for normality (Shapiro-Wilk, p = 0.297 and p = 0.614, respectively) and failed tests for equal variance (Fisher's F test, p < 0.05). A Mann-Whitney

200    Rank Sum test indicated that the δ$^{18}$O isotopes in precipitation were not different for summer (median = -8.3 ‰) and winter (median = -7.4 ‰, U = 39, p = 0.061). For δD, the summer had a significantly lower median value (median = -64.3 ‰) than winter (median = -46.5 ‰, U = 22, p = 0.004). A one-way ANOVA found no significant difference

among sites in $\delta^{18}O$ (F $_{(3,43)}$ = 0.323, p = 0.809) or $\delta D$ isotopes (F $_{(3,43)}$ = 0.361, p = 0.785) in precipitation when samples from all months and sites were included ($n$ = 47).

**3.5 $\delta^{13}C$ and $\delta^{18}O$ of α-cellulose – temperature, humidity, precipitation**

Across all sites, measurements of monthly average leaf $\delta^{18}O$ and $\delta^{13}C$ values for α-cellulose extracted from leaf tissues (hereafter $\delta^{18}O_{leaf}$ and $\delta^{13}C_{leaf}$) had a strong positive correlation (Pearson's r = 0.877, p < 0.001, $n$ = 24; Fig. 5) and segregation between winter and summer values (p < 0.001). The $\delta^{18}O_{leaf}$ or $\delta^{18}O_{root}$ did not correlate with $\delta^{18}O$ in precipitation across all sites (Table 1).

[revised manuscript text omitted]

We did not find a relationship between $\delta^{18}O$ in precipitation and the $\delta^{18}O_{leaf}$ in our one-year study. It may be that because the $\delta^{18}O$ in leaf water is controlled by source water and humidity, any changes in humidity confound a direct relationship between source water $\delta^{18}O$ values and $\delta^{18}O$ of leaf water and $\delta^{18}O_{leaf}$. Still, the $\delta^{18}O$ in plant water pools and $\delta^{18}O_{leaf}$ are primarily influenced by $\delta^{18}O$ in precipitation. It may also be possible that the variation in source area did not affect the $\delta^{18}O$ enough for us to detect a significant impact on the $\delta^{18}O_{leaf}$. Based on this work, the source area could influence $\delta^{18}O_{leaf}$ in peat records if source area changes were greater than what we observed. Without experimentation we can not disentangle temperature and humidity using $\delta^{13}C$ and $\delta^{18}O$; further work is needed to understand the relationship between $\delta^{18}O_{root}$, $\delta^{18}O$ of precipitation, and that of root and leaf waters. Identifying sources of potential water would also add value, especially considering anecdotes of local differences in fog in the Falkland Islands, which may be an unappreciated source of water for *P. flabellata*. Across the geographic range in the South Atlantic, *P. flabellata* may record a larger latitudinal gradient of isotopes in precipitation, as well as temperature and humidity, than recorded in the Falkland Islands.

Despite this limitation, establishing the seasonal patterns recorded by *P. flabellata* cellulose in the Falkland Islands does enable us to test paleoclimate hypotheses regarding the climate dynamics in the South Atlantic (e.g. Turney et al., 2016) and Southern Hemisphere westerly wind behavior within the regions where *P. flabellata* occurs. As with any paleoenvironmental reconstruction, inferences are constrained by the temporal grain and resolution (Jackson, 2012). While *P. flabellata* is sensitive to inter-seasonal differences and forms highly productive peat records, we stress that paleoclimate reconstructions from *P. flabellata* peats will represent an integrated signal of broader climate trends, and not annual-scale or seasonal records. Our calibration study is based on a modern snapshot of environmental conditions influencing $\delta^{13}C_{leaf}$ and $\delta^{18}O_{leaf}$ values in *P. flabellata*, so investigators conducting downcore peat reconstructions using $\delta^{18}O_{leaf}$ must consider that changes in $\delta^{18}O_{leaf}$ are modified by precipitation source and changes in humidity through effects on stomatal conductance. We recommend measuring $\delta^{13}C_{leaf}$ and $\delta^{18}O_{leaf}$ of subsamples with comparable time-averaging (leaf, root, bulk subfossils; Fig. 8) by incorporating a large sampling of multiple leaf fragments ($n \geq 10$) from each horizon (e.g. 1 cm intervals), and to interpret this signal to integrate multiple years (as determined by the sediment accumulation rate; Fig. 8). Time-averaging within a given 1 cm horizon of even highly productive peat records with sedimentation rates ~ 20 to 30 yr cm$^{-1}$ would be greater than the age of an individual tussac leaf.

Peat-based reconstructions may be limited to identifying periods of warm/dry or cold/wet conditions that are similar to (or more extreme than) the observed seasonal variations we report here. Thus, we believe that $\delta^{13}C_{leaf}$ and $\delta^{18}O_{leaf}$ time-averaged values can reliably be used to identify the timing of transitions between warmer and drier conditions to colder and wetter conditions that correlate with major hemispheric drivers in climate. Resolving temperature and moisture signals independently would likely require growth chamber or warming experiments, which was beyond the scope of this study, but which could help develop this proxy further. For now, this proxy remains a qualitative indicator,

though it has potential to become a quantitative reconstruction if evaluated via experimentation or in tandem with other plant or microbial biomarkers.

[revised manuscript text omitted]

545 **Figure 8. a)** Conceptual model of how climate variation influences biomass $\delta^{13}$C and $\delta^{18}$O values in *P. flabellata* through effects on stomatal conductance and $\delta^{18}$O of precipitation. **b)** Seasonal shifts in leaf $\delta^{13}$C and $\delta^{18}$O, with open circles representing winter (cool/wet conditions) and closed circles representing summer (warm/dry conditions). **c)** Diagram of a proposed paleoclimate reconstruction workflow and interpretation of time-averaged (interannual) measurements of $\delta^{13}$C and $\delta^{18}$O in peat macrofossils.